# Mating system manipulation and the evolution of sex-biased gene expression in *Drosophila*

Paris Veltsos [1,2], Yongxiang Fang[3], Andrew R. Cossins[3], Rhonda R. Snook [4,5] & Michael G. Ritchie [1]

Sex differences in dioecious animals are pervasive and result from gene expression differences. Elevated sexual selection has been predicted to increase the number and expression of male-biased genes, and experimentally imposing monogamy on *Drosophila melanogaster* has led to a relative feminisation of the transcriptome. Here, we test this hypothesis further by subjecting another polyandrous species, *D. pseudoobscura*, to 150 generations of experimental monogamy or elevated polyandry. We find that sex-biased genes do change in expression but, contrary to predictions, there is usually masculinisation of the transcriptome under monogamy, although this depends on tissue and sex. We also identify and describe gene expression changes following courtship experience. Courtship often influences gene expression, including patterns in sex-biased gene expression. Our results confirm that mating system manipulation disproportionately influences sex-biased gene expression but show that the direction of change is dynamic and unpredictable.

[1] Centre for Biological Diversity, School of Biology, University of St Andrews, Fife, St Andrews KY16 9TH, UK. [2] Department of Ecology and Evolution, University of Lausanne, 1015 Lausanne, Switzerland. [3] Centre for Genomic Researc, Institute for Integrative Biology, University of Liverpool, Liverpool L69 7BX, UK. [4] Department of Animal & Plant Sciences, University of Sheffield, Sheffield S10 2TN, UK. [5] Zoologiska Institutionen (Ekologi), Stockholm University, 106 91 Stockholm, Sweden. Correspondence and requests for materials should be addressed to R.R.S. (email: rhonda.snook@zoologi.su.se) or to M.G.R. (email: mgr@st-andrews.ac.uk)

The majority of genes show sex-biased expression variation in dioecious animals[1–3]; for example, around three quarters of the genes of *Drosophila melanogaster* are expressed differently in males and females[1]. Many behavioural, physiological and morphological differences between the sexes result from these differences in gene expression, but the evolutionary forces leading to the observed current balance of sex bias in expression, and how stable or labile this balance is, are largely unknown[2, 3]. Sexually antagonistic selection is thought to play a major role in contemporary levels of sex-biased expression, such that the balance reflects the relative strength of expression optima in the two sexes[4]. Genes that show male sex-biased expression evolve rapidly between species[5] and populations[6]. Comparisons across bird species with mating system variation show that high phenotypic sexual dimorphism is associated with higher levels of turnover of male-biased gene expression[7]. Within-species comparisons show that subordinate male turkeys have a more feminised transcriptome (i.e. a greater level of expression of female-biased genes)[8]. Such evolutionary patterns have led to the prediction that strong sexual selection on males can contribute to higher expression of male-biased genes (a masculinised transcriptome)[7–9].

A potential interaction between mating system and sex-biased gene expression variation was elegantly demonstrated with *Drosophila melanogaster*. Hollis et al.[9] subjected lines of the normally polyandrous *D. melanogaster* to experimental evolution under mating system manipulation, involving enforced monogamy

versus polygamy. After up to 100 generations of experimental evolution, virgin flies from monogamous lines showed, on average, a more feminised pattern of gene expression in the transcriptome of both heads and whole bodies in both sexes. Both female-biased genes were upregulated and male-biased genes downregulated under monogamy. Hollis et al.[9] concluded that, under a polyandrous mating system, sexual selection on males leads to increased expression of sexually antagonistic male-advantageous genes, but this source of selection is diminished under monogamy, leading to a change in sex-biased gene expression (in both sexes) towards a female optimum. This study demonstrated that manipulating the mating system leads to rapid changes in levels of sex-biased gene expression in the transcriptome, although no genes were significantly differentially expressed between the treatments. Most current evidence suggests that increased expression of male-biased genes will be favoured by stronger sexual selection, but sexual antagonism is expected to be dynamic due to intersexual coevolution, so details could be unpredictable[10, 11]. Moreover, optimal levels of sex-biased gene expression will likely be tissue-specific or context-specific, with some genes known to show rapid changes in expression such as those responding to social context, courtship or mating[12–15].

Here, we report a dramatically different outcome to a similar experiment as Hollis et al[9]. We subjected a different naturally polygamous species, *Drosophila pseudoobscura*, to experimental evolution under enforced monogamy and elevated polyandry for more than 150 generations. We quantified changes in gene

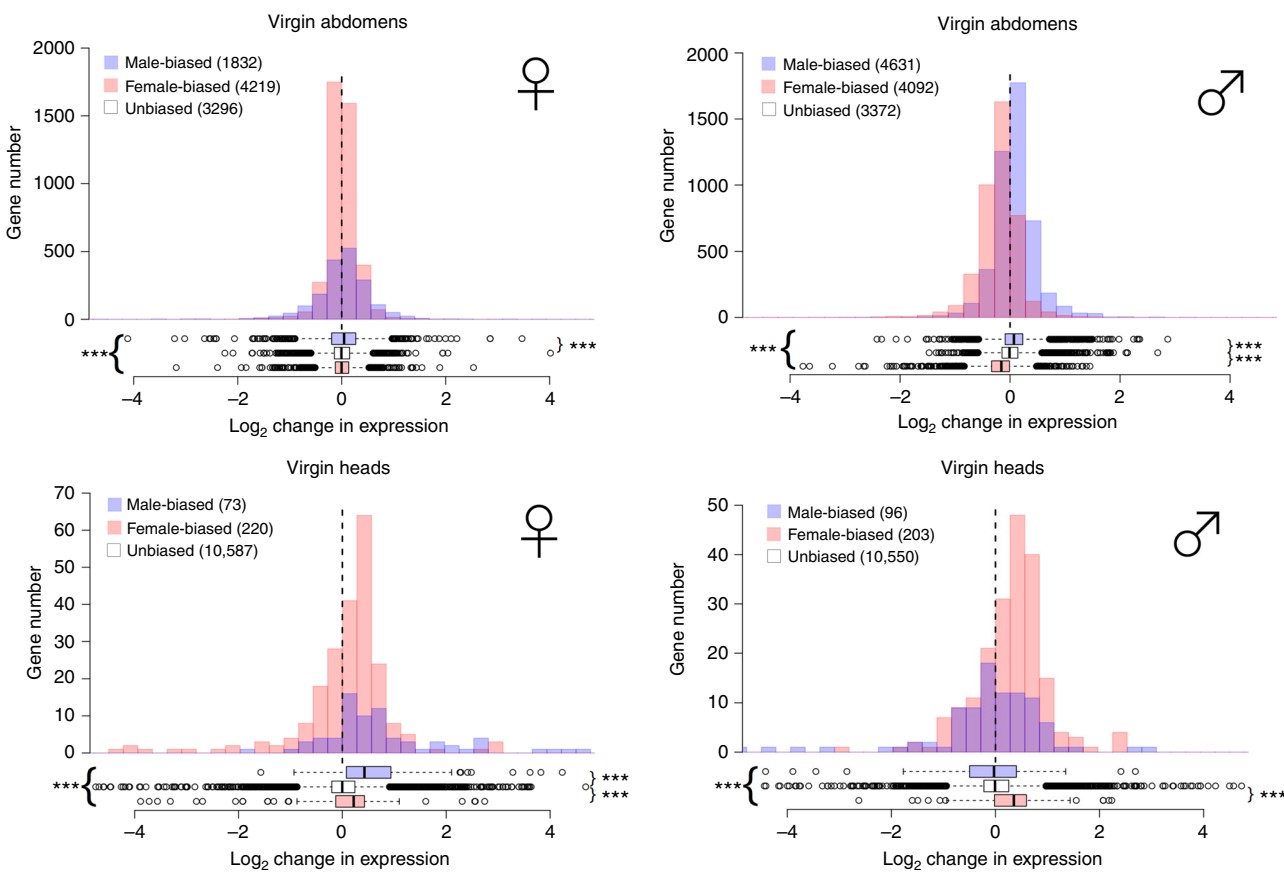

**Fig. 1** Gene expression changes following experimental mating system manipulation in virgin flies. Positive values on the *x* axis correspond to higher expression under Monogamy, negative values to higher expression under increased Polyandry. Colours indicate male-biased (blue), female-biased (pink) and unbiased (white) genes in abdomens (top) and heads (bottom). For clarity, unbiased genes are omitted from the histogram. The significance level of Mann–Whitney rank tests on the average level of sex-biased expression is indicated by asterisks (*** < 0.001). Asterisks to the right of the box plots summarise comparisons of male-biased and female-biased genes with unbiased genes, and those to the left between male-biased and female-biased genes

**Table 1 Summary of $\chi^2$ tests on 10% FDR DE genes in each of the eight possible sexual selection contrasts combining tissue, sex and courtship bias**

| Tissue | Gene type | No. genes | No. FDR < 10% | Proportion | $\chi^2$ c/f unbiased |
|---|---|---|---|---|---|
| Virgin female abdomens | Male-biased | 1832 | 4 | 0.0022 | NS |
| | Female-biased | 4219 | 5 | 0.0012 | NS |
| | Unbiased | 3296 | 3 | 0.0009 | |
| Virgin male abdomens | Male-biased | 4631 | 11 | 0.0024 | NS |
| | Female-biased | 4092 | 2 | 0.0005 | NS |
| | Unbiased | 3372 | 5 | 0.0015 | |
| Virgin female heads | Male-biased | 73 | 8 | 0.1096 | 152.2, <0.0001 |
| | Female-biased | 220 | 1 | 0.0045 | NS |
| | Unbiased | 10,587 | 43 | 0.0041 | |
| Virgin male heads | Male-biased | 96 | 7 | 0.0729 | 44.9, $p < 0.0001$ |
| | Female-biased | 203 | 3 | 0.0148 | NS |
| | Unbiased | 10,550 | 71 | 0.0067 | |
| Courted female abdomens | Male-biased | 5134 | 6 | 0.0012 | NS |
| | Female-biased | 3926 | 4 | 0.0010 | NS |
| | Unbiased | 3463 | 3 | 0.0009 | |
| Courted male abdomens | Male-biased | 5342 | 82 | 0.0154 | 2.98, $p = 0.083$ |
| | Female-biased | 3674 | 14 | 0.0038 | 11.44, $p = 0.0007$ |
| | Unbiased | 3073 | 33 | 0.0107 | |
| Courted female heads | Male-biased | 197 | 9 | 0.0457 | 8.2, $p = 0.0042$ |
| | Female-biased | 105 | 3 | 0.0286 | NS |
| | Unbiased | 10,693 | 171 | 0.0160 | |
| Courted male heads | Male-biased | 216 | 7 | 0.0324 | 6.11, $p = 0.013$ |
| | Female-biased | 93 | 0 | 0.0000 | |
| | Unbiased | 10,608 | 120 | 0.0113 | |

expression in heads and abdomens of both virgin and courted flies of both sexes and compared expression changes for male-biased, female-biased and unbiased genes. Sex bias was defined from the same baseline population used to establish the selection lines, maintained at an equal sex ratio. Some of our results replicate the prediction of increased feminisation under monogamy arising from Hollis et al.[9], but the majority do not, with masculinisation of the transcriptome under monogamy being the most consistent pattern found. Changing the biological context (courtship status of the flies) in which gene expression was measured also influenced these patterns. For example, the transcriptome of male abdomens showed masculinisation under monogamy in virgins, but became more feminised following courtship. Our results demonstrate that changes in the expression patterns of sex-biased genes, which occur as a consequence of mating system manipulation, are context-dependent, dynamic and hard to predict. The response of sex-biased gene expression to sexual selection intensity may differ even between similar polyandrous species.

## Results

**Mating system manipulation and gene expression.** After 157 generations of experimental evolution under either enforced monogamy (M: one male housed with one female) or elevated polyandry (E: one female housed with six males) we used RNAseq to quantify gene expression variation in both sexes. We dissected abdomens and heads (Hollis et al.[9] examined whole bodies and heads) before RNA extraction and examined both virgins and individuals collected immediately after successful courtship (within 10 s of mounting, which is prior to sperm transfer[16]). Expression data differed substantially between tissue type and sex, and were therefore analysed separately for each sex, courtship status and tissue type, except when calling sex-biased gene expression, when male and female data were combined. Sex bias was called from an independent set of 'Baseline' flies reared under mass culture with an equal sex ratio (the typical mating system of

this species involves each female mating with two or three males[17]). Differential expression was called using EdgeR[18] and there were 111/223 (virgin/courted) and 5122/6007 (virgin/courted) male-biased genes and 252/151 (virgin/courted) and 4422/3926 (virgin/courted) female-biased genes identified in heads and abdomens, respectively. We tested how manipulation of sexual selection intensity influenced gene expression, particularly how unbiased and sex-biased genes changed expression according to sex, tissue and courtship status. A more 'gendered' transcriptome would arise if there is more pronounced upregulation of sex-biased genes of one type. For example, masculinisation of the transcriptome occurs when male-biased genes are upregulated significantly more than unbiased or female-biased genes.

The most comparable analysis to the whole body analysis of Hollis et al.[9], is a comparison of overall gene expression changes in virgin male and female abdomens. We expect sex-biased gene expression to be most divergent in this body segment given that they house the sex-specific reproductive tissues. In the abdomens of virgin females, male-biased genes increased expression more than unbiased genes under monogamy, leading to masculinisation (Fig. 1, top). In virgin male abdomens female-biased genes were, on average, downregulated under monogamy and male-biased genes were upregulated, giving a clearer signal of masculinisation under monogamy (Fig. 1, top). We also quantified changes in gene expression in heads (Fig. 1, bottom) where sex-specific tissue differences are less pronounced (fewer genes show sex-biased expression). We found higher expression of both male-biased and female-biased genes under monogamy in females; however, the increase of expression was greater for male-biased than for female-biased genes, so the net effect was masculinisation of the transcriptome. In male heads, the greatest response under monogamy was an increase in the expression of female-biased genes, so the transcriptome was feminised. Hence, overall, virgins showed tissue-specific responses, with feminisation of male heads, but masculinisation of male abdomens and both female heads and abdomens under monogamy. We

performed bootstrapping analysis on the changes in gene expression as a further test of whether changes in gene expression were biased in one direction; these gave the same results as those illustrated in Fig. 1 (Supplementary Table 1).

To assess the extent to which genes expressed in sex-specific gonadal tissue may be influencing the patterns found in abdomens, we repeated the analyses after removing genes likely to be highly expressed in gonads. First, we used published data on genes expressed in *D. pseudoobscura* testes and ovaries (ref: data set GSE52058) to remove genes from our abdominal data set that differ in expression between the male and female reproductive tissues and repeated the analysis above. While the magnitude of change is reduced, the direction of change towards masculinisation remains the same (Supplementary Fig. 1). Second, we repeated the analysis after removing sex-specific genes (i.e. any expressed only in either male or female samples in our data), which would include gonad-specific genes. The results also remained largely unchanged, with one exception that virgin female abdomens showed weak masculinisation under monogamy (Supplementary Fig. 2). We can therefore conclude that the major patterns of masculinisation versus feminisation are not primarily driven by changes in these sex-specific tissues.

Most of the changes in gene expression as a consequence of mating system manipulation are relatively small, and the patterns described above are based on differences across all loci. We repeated the analysis considering only genes that are significantly differentially expressed by treatment, which allows us to ask whether either type of sex-biased loci are more likely to respond to the manipulation of mating system. Hollis et al.[9] did not report such an analysis, due to lack of differentially expressed genes. We present results from an FDR of <10% to balance stringency and gene number, but the results were qualitatively the same with a 5% FDR threshold. For each contrast, we tested whether differentially expressed (DE) genes were disproportionately composed of sex-biased genes (Table 1). Male-biased genes were more likely to respond to the treatment than unbiased genes (the only case in which female-biased genes responded differently, they were less likely to respond). These results were not driven by an extreme replicate within a treatment given that heatmaps of expression changes in these DE genes are largely consistent across replicates (Supplementary Fig. 3).

The results shown in Fig. 1 and Supplementary Fig. 1 follow the analytical procedure of Hollis et al.[9], which compared the mean expression change across groups of genes classified by their sex bias. Such a statistical approach can be criticised for potentially inflating degrees of freedom, since the genes within each sex bias class are unlikely to change expression independently of each other, and such tests may be biased by outliers. To offset such concerns, we also present an analysis that uses randomisation to avoid estimating the degrees of freedom, and a measure that incorporates gene level variation, so that both the expression level of a gene and the confidence in the observed value were taken into account. We call the measure the 'standardised logFC' (stdLogFC) and it is effectively an adjusted

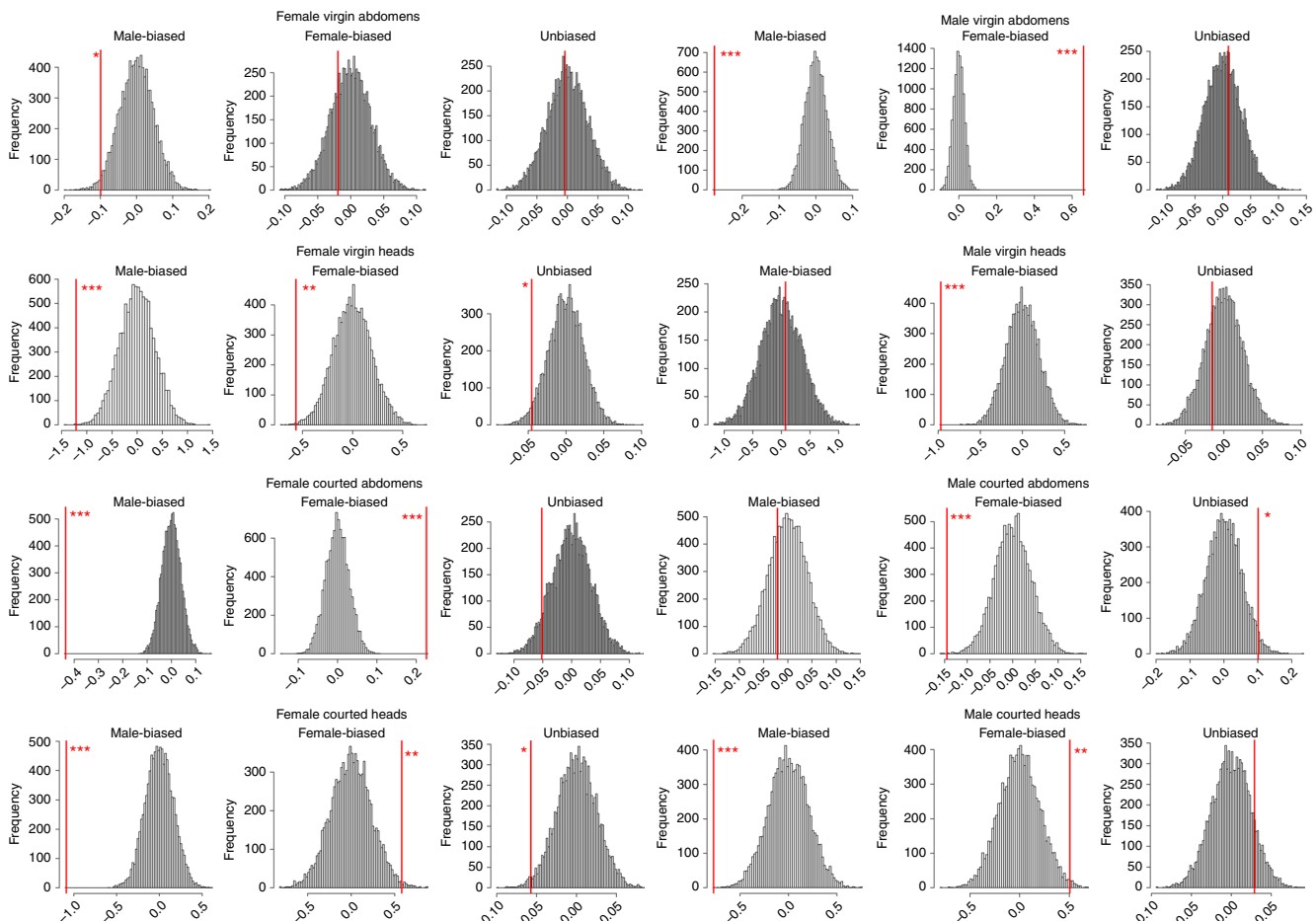

**Fig. 2** Results of the balanced randomisation analysis of gene expression changes. Histograms summarise 10,000 iterations of switching the sign of stdLogFC. Red bars indicate the observed mean stdLogFC. Positive values in the x axis indicate higher expression under Polygamy. Stars indicate the proportion of iterations with a more extreme value than the observed (*** < 0.001, ** < 0.01, * < 0.05)

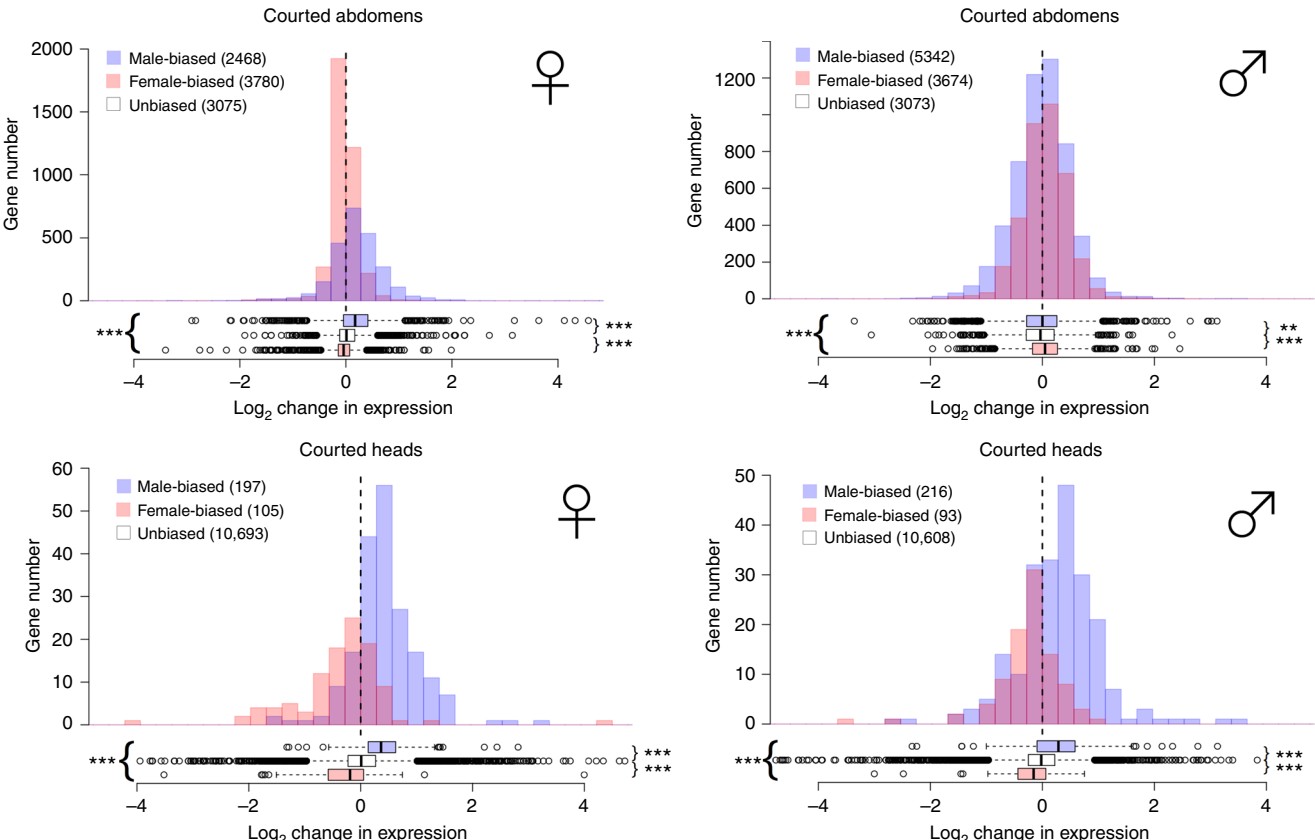

**Fig. 3** Gene expression changes following mating system manipulation in courted flies. Positive values on the x axis correspond to high expression under Monogamy, negative values to high expression under increased Polyandry. Colours indicate male-biased (blue), female-biased (pink) and unbiased (white) genes in abdomens (top) and heads (bottom). For clarity, unbiased genes are omitted from the histogram. The significance level of Mann–Whitney rank tests on the average level of sex-biased expression is indicated by asterisks (*** < 0.001, ** < 0.01). Asterisks to the right of the box plots summarise comparisons of male-biased and female-biased genes with unbiased genes, and those to the left between male-biased and female-biased genes

p-value from edgeR with a sign applied to indicate which treatment has the higher expression, with E > M being positive (Methods). Randomisation was of the direction of changes in gene expression with respect to sexual selection treatment. We call this 'balanced randomisation', because it overcomes any inflation of variation that ordinary randomisation would produce by randomising equal numbers of samples from the two treatments. Our test statistic was obtained by comparing the average stdLogFC for each sex-biased gene set to that obtained by resampling (with 10,000 iterations) each gene set, after randomly inverting the sign for half of the genes. This inversion randomises the influence of the sexual selection treatment. Comparing the observed and resampled distributions indicated the likelihood of obtaining our results by chance, and confirms the analyses reported above (Fig. 2).

We also used an alternative randomisation approach, randomising individual replicates (libraries). This is theoretically superior to the 'balanced randomisation' across genes because it does not change any covariance structure possessed by the data set and accounts for outlier libraries that may disproportionately influence the result. However, this analysis is less than ideal because of our limited number of replicates (Methods). Nevertheless, the results support those of the 'balanced randomisation' (Supplementary Fig. 4).

**Courtship and gene expression**. Patterns of gene expression can change due to social encounters, including courtship experience, so we investigated the effect of both courtship experience and

mating system manipulation by repeating our previous analysis on tissues from individuals which had experienced courtship (Fig. 3). In some cases, the direction of changes evolving under monogamy was different from that seen in virgins, with respect to sex-biased gene expression. For example, gene expression in male heads was masculinised under monogamy, while in virgins it was feminised. The patterns also changed in male abdomens. Again, these patterns were all confirmed by bootstrapping analyses of the changes in gene expression (Supplementary Table 1) and the resampling analyses (Fig. 2; Supplementary Fig. 4).

To examine the nature of those genes responding to courtship in more detail we used the technically replicated Baseline individuals to identify genes that were differentially expressed between virgin and courted flies. The number of such genes differed between sexes, tissues and by sex-bias status (Table 2). Most were found in male heads. Perhaps not surprisingly, because heads do not have many sex-biased genes, the majority (≈87%) were not sex-biased. The pattern was very different for female abdomens, where the majority were male-biased genes (down-regulated in females), though over 50 female-biased genes were upregulated following courtship. Few genes showed differential expression between virgin and courted Baseline flies in female heads and male abdomens (Table 2).

Figure 4 is a summary of the major patterns of masculinisation and feminisation seen across all comparisons between sexes, tissues, and courtship status. All analyses (mean logFC difference, mean stdLogFC from balanced randomisation, mean stdLogFC from ordinary randomisation) resulted in the same qualitative conclusions, and our results clearly do not support the prediction

**Table 2 Number of genes which are differentially expressed (FDR 10%) as a result of courtship experience in flies from the Baseline population, and their sex-bias status**

| | Female heads | | | | Male heads | | | | Female abdomens | | | | Male abdomens | | | |
|---|---|---|---|---|---|---|---|---|---|---|---|---|---|---|---|---|
| | total | ♂ | unbiased | ♀ | total | ♂ | unbiased | ♀ | total | ♂ | unbiased | ♀ | total | ♂ | unbiased | ♀ |
| Total DE | 16 | 3 | 11 | 2 | 748 | 35 | 656 | 57 | 321 | 206 | 36 | 79 | 1 | 0 | 1 | 1 |
| Up in virgin | 12 | 1 | 10 | 1 | 273 | 2 | 259 | 12 | 245 | 199 | 26 | 20 | 0 | 0 | 0 | 0 |
| Up in courted | 4 | 2 | 1 | 1 | 475 | 33 | 397 | 45 | 76 | 7 | 10 | 59 | 1 | 0 | 1 | 1 |

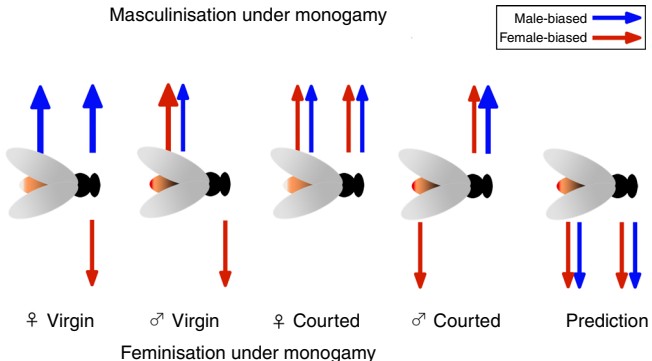

Masculinisation under monogamy

Male-biased →
Female-biased →

♀ Virgin　♂ Virgin　♀ Courted　♂ Courted　Prediction

Feminisation under monogamy

**Fig. 4** Summary of the direction of change of sex-biased gene expression in heads and abdomens. The direction of the arrows indicates whether there is masculinisation (up) or feminisation (down), under monogamy. The colour of the arrows represents male (blue) or female (red) biased genes. Only cases when the means between sex-biased gene subsets differ significantly are represented by arrows, and thicker arrows indicate greater mean logFC, within the contrast. Most results do not conform to the prediction of masculinisation under monogamy (farthest right)

that monogamy will result in feminisation of gene expression in both males and females.

**Functional analysis of differentially expressed genes.** To examine the functional classes of genes showing differential expression in our study we completed GO term enrichment analyses of those genes which were differentially expressed between mating system treatments (in the experimentally evolved flies) and flies of different courtship experience (in the Baseline flies).

Supplementary Data 1 presents those gene functions which responded to the manipulation of mating system. Interestingly, glutathione metabolism terms were enriched in three of the eight contrasts (female heads and virgin male abdomens). Glutathione metabolism has been linked to mating success in *D. melanogaster*[19], where glutathione transferases have higher expression in males displaying greater competitive reproductive success. The oxidative function of glutathione metabolism may contribute to condition-dependent expression of sexually dimorphic signals in male birds[20]. Other notable terms found include post-mating behaviour and insemination for genes expressed in virgin female heads, antimicrobial defence in polyandrous male virgin heads, and odorant binding in male courted heads. The term 'establishment or maintenance of polarity of embryonic epithelium' was found in both virgin and courted female abdomens, and is perhaps associated with the higher fecundity of females from the polyandrous lines[21].

Supplementary Data 2 presents those gene functions over-represented amongst those genes changing expression following courtship experience. Those in female abdomens included 'immunity' related and 'pheromone binding' functions. For male heads there were terms associated with learning and memory

such as 'olfactory learning' and 'rhodopsin-associated signalling', and in both sexes mitochondrial-related terms were enriched. McGraw et al.[22] identified genes responding to mating and ejaculate components in *D. melanogaster* and we can ask if similar genes were involved. We found some overlap in the genes identified in both studies, but not significantly more than would be expected by chance (12.5% compared to 9.8%; $\chi^2 = 2.54$, d$f = 1$, $p = 0.11$). Those genes showing a common association with courtship and mating between the two species were primarily associated with immune function (9/40), methyltransferase and endopeptidase activities.

**Correlations between sexes and species.** In order to examine the extent to which gene expression changes due to mating system manipulation were consistent between the sexes, we correlated changes in gene expression for those which showed significant differential expression. Expression changes were correlated between the sexes, except in virgin abdomens (Table 3). Hence the sexes did not respond completely independently, suggesting there may be constraints on the ability of genes to show sex-specific responses to mating system manipulation (as proposed by Hollis et al.[9]). Bootstrapping these correlation coefficients did not suggest they differed between the treatments as might have been expected if, for example, any such constraints were reduced under monogamy (Table 3).

We also directly compared the difference in expression between high and low sexual selection treatments from our study with those in Hollis et al.[9] for the genes with clear homologues in *D. melanogaster*, as identified in Flybase. The data from heads are comparable because the two studies used the same tissue and methodology (RNAseq). There is no evidence that the genes responding to sexual selection were diversifying more rapidly between species, because the (DE) genes significantly responding to mating system manipulation in *D. pseudoobscura* were more likely than expected by chance to have homologues in *D. melanogaster* (390/450), compared to non-responding genes (11,616/15,899; $\chi^2 = 40.38$, d$f = 1$, $p < 0.0001$). There was some evidence of consistency in which genes changed expression between the species, because changes across all genes were correlated, positively in female heads and negatively (but with a very small correlation coefficient) in male heads (Supplementary Fig. 5). This negative correlation could reinforce our conclusion that the nature of sex-specific selection differs between the species, but needs to be taken with caution as these correlations are not significant when only calculated with genes which were differentially expressed in our study, and the direction of change in male heads is small and influenced by outliers.

**Chromosomal distribution of DE genes.** The evolutionary dynamics of sex-linked and autosomal genes often differ, especially for genes under sex-specific or antagonistic selection[23]. We performed $\chi^2$ tests to test the effect of chromosomal location on genes responding to sexual selection and on sex-biased genes. The DE genes for any contrast in each of those categories were grouped together. There was a small enrichment of genes

**Table 3 Correlation analysis of gene expression changes between the sexes, with bootstrapped 95% confidence intervals**

|  | Elevated polyandry | | | Monogamy | | |
|---|---|---|---|---|---|---|
|  | Mean | s.e. | 95% CI | Mean | s.e. | 95% CI |
| Virgin abdomens | 0.209 | 0.304 | (−0.237 to +0.759) | 0.189 | 0.334 | (−0.290 to +0.826) |
| Virgin heads | 0.935 | 0.035 | (0.866 to 0.990) | 0.976 | 0.014 | (0.944 to 0.976) |
| Courted abdomens | 0.655 | 0.095 | (0.457 to 0.846) | 0.587 | 0.078 | (0.448 to 0.752) |
| Courted heads | 0.922 | 0.034 | (0.854 to 0.957) | 0.952 | 0.022 | (0.906 to 0.982) |

**Table 4 Chromosomal distribution of genes which are DE (FDR 10%) following mating system manipulation (679 genes), and by sex (11845 genes), in any contrast**

|  | Chromosome | DE on focus chromosome | Remaining DE | $\chi^2$ | p |
|---|---|---|---|---|---|
| DE by sexual selection treatment | 2 | 144 (124) | 365 (385) | 4.45 | 0.03 |
|  | 3 | 110 (99) | 399 (410) | 1.65 | 0.2 |
|  | 4 | 85 (99) | 424 (410) | 2.52 | 0.11 |
|  | XR | 97 (99) | 412 (410) | 0.04 | 0.84 |
|  | XL | 73 (89) | 436 (420) | 3.45 | 0.06 |
| DE by sex | 2 | 2829 (2779) | 8523 (8573) | 1.2 | 0.27 |
|  | 3 | 2182 (2186) | 9170 (9166) | 0.01 | 0.92 |
|  | 4 | 2315 (2262) | 9037 (9090) | 1.53 | 0.22 |
|  | XR | 2054 (2140) | 9298 (9212) | 4.23 | 0.04 |
|  | XL | 1972 (1985) | 9380 (9367) | 0.11 | 0.74 |

The total number of genes assigned to Muller elements is 15,120. The expected numbers of genes, given the total in each chromosome, are indicated in parentheses.

responding to sexual selection on chromosome 2 ($\chi^2 = 4.45$, df = 1, $p = 0.03$, Table 4) and a slight deficit from XL ($\chi^2 = 3.45$, df = 1, $p = 0.06$, Table 4). In *D. pseudoobscura* XR has more recently become sex-linked (8–12 mya[24]) due to a translocation event. Both the X chromosome overall ($\chi^2 = 3.72$, $p = 0.05$) and the XR in particular ($\chi^2 = 4.23$, df = 1, $p = 0.04$, Table 4), show a deficit of sex-biased genes. A reduction of male-biased genes on the *D. pseudoobscura* X, stronger for XR, has been described before[1, 25]. Hence our data provide no evidence that sex-biased expression is evolving more quickly on either arm of the X chromosome or that sex-linked genes are more likely to respond to mating system manipulation.

## Discussion

Different gene expression optima in the two sexes underlie the evolution of sex-biased gene expression. Sex-limited gene expression may resolve intra-locus conflict, and is likely in sex-specific tissues such as testes and ovaries, especially for male-biased genes, which seem to be less pleiotropic than female-biased genes[1]. However, inter-locus conflict is extensive and, given that most genes are expressed in both sexes, the level of expression observed is thought to be a balance between the optimal levels for each sex[4]. Previously Hollis et al.[9] showed that the transcriptome of *D. melanogaster* males and females, in both whole bodies and isolated head tissue, became more feminised during experimental evolution under monogamy (although there was no differential expression between treatments). This seems to confirm the general pattern that male-biased gene expression evolves rapidly, probably in response to strong sexual selection. A previous microarray study of female *D. pseudoobscura* implied that sex-biased genes changed expression in monogamous females[26]. Here we use RNAseq to examine both sexes of *D. pseudoobscura* extensively and find that sex-biased genes are more likely to respond to mating system manipulation, but the direction of the response depends on the tissue type, sex and courtship status. Intriguingly, courtship results in predominantly male-biased genes changing expression in female abdomens and there is an overall masculinisation of the *D. pseudoobscura* transcriptome under monogamy (i.e. male-biased genes disproportionally increase, and female-biased decrease, in expression; Fig. 3).

Sexually antagonistic selection is expected to be dynamic because of constant coevolution between males and females, which would be reflected in a changing balance between male-favoured and female-favoured levels of gene expression[10, 11]. For example, different levels of male harm and female resistance evolved in populations of *D. melanogaster* adapted to different laboratory environments[27]. Nevertheless, sexual selection is expected to be stronger on males because they are subject to more intrasexual selection leading to a greater variance in reproductive success, which may be reflected at the level of gene expression. Indeed male-biased genes experience both a faster evolutionary turnover when under strong sexual selection intensity[7] and are more likely to diverge between species[5]. Studies of expression variation amongst lines of *D. melanogaster* have found rapid evolution of sex-biased expression, but more variance amongst lines in male-biased genes[6].

However, polyandry can benefit females[28] and seems to do so in our experimentally evolved populations: polyandrous females have higher fecundity and offspring hatching success compared to monogamous females[21]. These female benefits are likely to be associated with higher expression of female-biased genes in polyandrous females[26]. Imposing monogamy on female *D. melanogaster* also leads to a decrease in fecundity[29] and laboratory evolution of increased mating rate leads to a reduction in lifespan associated with many changes in gene expression[30]. Given that the natural mating system of both *D. melanogaster* and *D. pseudoobscura* is polygamy, experimental enforcement of monogamy is a relatively dramatic change to both species, suggesting that the optimum for female-biased genes is not necessarily similar to monogamous conditions. It is therefore possible that the evolutionary response to monogamy could involve larger changes to the optimum expression level of female-biased genes, compared to those of male-biased genes, contrary to the suggestion of Hollis et al.[9] (their Fig. 1). Such changes would cause masculinisation under monogamy, as we find here. That mating

system can act independently (at the phenotypic level), and in unexpected ways, on male and female optima has recently been demonstrated in passerine birds, where plumage dimorphism has been shown to increase under sexual selection. Surprisingly this seems to be because of stronger selection on females to be less colourful rather than increased selection on males to be more colourful[31]. Sex-specific selection may be unpredictable, and is almost certain to vary between tissues and classes of genes.

It is impossible to conclude with any certainty what differences between our study and that of Hollis et al.[9] are responsible for the contrasting experimental outcomes, both because the precise nature of sex-specific selection in either study species are unknown (and may differ) and because the experiments differed in aspects other than the study species which may have influenced the evolutionary response. One difference was the nature of the source population; we used a relatively recently collected, large outbred population to start our experiment whereas Hollis et al.[9] used an older stock that had been subjected to mutagenesis. Thus, different sources of genetic variation (standing genetic variation for *D. pseudoobscura* and induced mutations for *D. melanogaster*) were potentially available for selection to act on. Spontaneous mutations in *D. melanogaster* can have a larger effect on male than female fitness[32] and the directional effects of sexual selection can differ according to the nature of genetic variation segregating in populations[33]. Some of the differences between our results could therefore reflect differences in the nature of the available genetic variation.

Another potentially significant difference in design between the studies is that the high sexual selection treatment differed; in *D. pseudoobscura*, a polyandrous (one female, multiple males) environment was imposed whereas in *D. melanogaster* there was polygynandry (multiple females were exposed to multiple males). Therefore, the intensity and nature of sex-specific selection may differ in subtle but important ways. Indeed, *D. melanogaster* males show different gene expression changes depending on the structure of their social environment[14, 34]. In the manipulated *D. pseudoobscura* populations, the greater courtship intensity in polyandrous lines[35] could mean that *D. pseudoobscura* females were under stronger selection to resist courtship attempts from multiple males and correspondingly males may have been under more intense intrasexual selection. Males from the polyandrous lines exhibit elevated courtship frequencies[35], outcompete monogamy line males[36], and more severely negatively impact female fitness when mated to monogamous females[37]. While these responses suggest that strong selection has influenced males, this in turn will affect female fitness, and select for responses in females, potentially increasing the expression of female-biased genes.

A further difference between studies is that we quantified gene expression in heads and abdomens whereas Hollis et al.[9] studied heads and whole bodies. Gene expression comparisons have been criticised based on potential allometric differences of sex-specific tissues that could generate apparent treatment or species-specific differences independently of gene expression variation[38]. For allometry to explain why *D. pseudoobscura* primarily shows masculinisation under monogamy there should have been consistent changes in male-specific tissues (in both abdomens and heads) of *D. pseudoobscura* evolving under monogamy and the same body parts should have more female-specific tissues in *D. melanogaster*. This pattern seems rather unlikely, especially as we demonstrated that removing genes primarily expressed in the gonads does not alter the overall patterns. In addition, while allometric changes can confound interpretations of biases in transcriptomics (e.g. ref. [7]) the observed changes in expression following courtship experience demonstrate that allometric

changes in tissues cannot underlie all the patterns that we identify.

A final difference between the studies is the way sex bias was called. We called sex bias separately for each tissue and courtship status, using four technical replicates of a large population of the same strain as the experimental evolution lines were derived from, whereas Hollis et al.[9] used an external data set. While this may have introduced errors in identifying sex-biased gene expression in their study (because sex-bias evolves rapidly), it seems unlikely to produce a systematic bias and therefore cannot explain variation in results between studies. Homology analyses showed that genes which responded to sexual selection in *D. pseudoobscura* were not species-specific so the differences between the studies are unlikely to be driven by the rapid divergence of fast evolving species-specific genes. Instead, genes with homologues between the two species seem to have responded in a species-specific fashion, as their expression changes in the two studies were largely uncorrelated.

A notable result from our study is the extent to which courtship experience affects both sex-specific and tissue-specific changes in gene expression, and that these context-dependent changes differ between the evolved lines. To ensure complete courtship, we allowed pairs to interact until intromission and then pairs were separated within 10 s. In this species, sperm transfer does not begin until at least 60 s into copulation, making it unlikely for males to have transferred any ejaculate to their partners[16]. Nevertheless, some seminal transfer may have occurred so we cannot rule out such effects. A few other studies have also demonstrated changes in gene expression underlying courtship behaviour[39] or following choice of mating partner[40]. Gene expression changes due to courtship may reflect strategic investment in costly mating signals when required. For example, male flies can vary cuticular hydrocarbons under different social conditions[14] or when experiencing (even only seeing) females[41, 42]. In *D. melanogaster* seminal products are very responsive to the social environment, which may determine the potential for intrasexual selection, especially sperm competition[43]. Little is known about the time scale of such gene expression changes during courtship. In females, courtship or copulation may lead to the upregulation of genes involved in oviposition, defence from male seminal products or sometimes quite precise immunological anticipation of potential sexually transmitted diseases[13, 22, 40, 44].

The genes that were differentially expressed in response to courtship in *D. pseudoobscura* differed strikingly between the sexes and tissues. In males most such genes were found in the head, consistent with the importance of behavioural interactions, though the majority were not sex-biased and of those that were, were equally likely to be male-biased or female-biased. In females, most genes affected by courtship were male-biased which decrease in expression after courtship (this is probably the opposite of what would be expected if these changes resulted from antagonistic sexual effects). Functional analyses of the genes showing differential expression here provide interesting results. Immune function genes differ between monogamous and polyandrous lines, and respond to courtship. These could arise due to an increased risk of mating-related infections or other defence-related responses in female abdomens (Supplementary Data 2). Immunological anticipation of deleterious consequences of mating has been reported in *Drosophila*[44]. Other functions that are easily interpretable are sensory processing genes changing expression with courtship and fecundity-related genes changing expression due to mating system manipulation. Some of the genes that changed expression following courtship were sex-biased (Table 2), and we found that the effect of mating system manipulation led to a more consistent masculinisation of the transcriptome under monogamy in courted flies (Fig. 2).

*Drosophila pseudoobscura* allows consideration of the effects of short-term and long-term sex linkage, because a chromosomal fusion has caused a previously autosomal chromosome arm to become part of the X chromosome (the *D. pseudoobscura* XL is homologous to the X in *D. melanogaster*, while the *D. pseudoobscura* XR is homologous to an autosomal region of *D. melanogaster*). This allows us to test if rates or patterns of gene expression evolution diverge between these regions. We found an almost significant deficit of genes responding to sexual selection on the older part of the X chromosome (XL) and that XR contains a deficit of sex-biased genes. These patterns may be influenced by demasculinisation of the X, since genes responding to sexual selection were often male-biased. Demasculinisation of the *D. pseudoobscura* X has been reported before and is thought to result from gene movement in and out of the X chromosome, although it is very difficult to disentangle confounding effects of dosage compensation in *Drosophila*[1, 25].

The imposition of monogamy on *D. pseudoobscura* results in a complex pattern of changes in gene expression, disproportionally involving changes in sex-biased genes. We find that male-biased genes often show increased expression under monogamy, especially following courtship experience, contrary to predictions that monogamy may lead to greater expression of female-biased genes. Masculinisation of the transcriptome is seen in both males and females, though the strength of the changes differs between heads and abdomens and with courtship experience. Monogamy only leads to an overall feminisation of the sex-biased transcriptome in virgin male heads and abdomens of courted males. The main effect of mating system manipulation in *D. pseudoobscura* is to masculinise, rather than feminise, the transcriptome under monogamy, providing a notable contrast with a previous study of *D. melanogaster* and at odds with the directional prediction arising from that study. We conclude that predicting changes in sex-biased gene expression evolution in response to sexual selection is not as straightforward as previously thought, and precise expectations may need to be tailored to specific species, genotypes, tissues and biological contexts.

## Methods

**Fly maintenance and preparation**. Line creation and maintenance is described elsewhere[21, 37]. Briefly, 50 wild caught females were collected from Tucson Arizona, reared in the lab for four generations and then used to establish the selection lines. Both the original stock (Baseline) and the subsequent selection lines were kept in standard fly medium at 22 °C on a 12 h day length. The selection regime was either monogamy (M), in which one female was housed with one male, or polyandry (E) where one female was housed with six males. The Baseline population was maintained under mass culture with an equal sex ratio. The sex ratio of the Baseline is 50% but females typically mate with two or three males[17]. The effective population size of the baseline population is well over 500 and is equalised between the treatments at around 120[45]. At each generation, offspring are collected and pooled together for each replicate line, and a random sample from this pool is used to constitute the next generation in the appropriate sex ratios, thus reflecting the differential offspring production across families.

Selection was relaxed prior to the experiment to reduce the likelihood of maternal effects. Parental flies were collected en masse until reproductively mature, then placed on egg laying plates. Plates were removed every 24 h, and after 48 h control density vials were established with 100 first instar larvae. All flies were placed in same sex groups of up to 12 individuals, then 25 samples per line, tissue, sex and courtship treatment were collected in 2 h periods. For the courtship treatment, females were aspirated into fresh food vials, and a male introduced after at least 1 min. Within 10 s of mounting, the pair was dropped in liquid nitrogen. For the virgin treatment, flies rested in individual food vials alone for at least 1 min before freezing and dissection. The encounter with a male was the only systematic difference between the two groups.

All dissections were performed within 5 days in a design as balanced as logistically possible, on 5-day-old flies. All flies were kept in four time-shifted incubators to avoid circadian effects on gene expression. Heads and abdomens from both sexes were dissected in 4 °C RNAlater (Sigma-Aldrich) and care was taken to include antennae. The samples were stored at 4 °C overnight, and then at −20 °C until RNA extraction.

RNA was extracted using Trizol plus (Ambion) with RNA purelink columns for cleanup and Dnase treatment. Ribozero (Epicentre) was used to remove rDNA.

RNA extractions were assessed for quality using Nanodrop (Thermo Scientific), Bioanalyser (Agilent) and Qubit (Life Technologies). RNAseq libraries were prepared from this material using the Epicentre ScriptSeq v2 RNAseq Library Preparation Kit following 15 cycles of amplification, and libraries purified using AMPure XP beads (Beckman Coulter Inc.). Each library was quantified using Qubit and the size distribution assessed using the Agilent 2100 Bioanalyser. These final libraries were pooled in equimolar ratios. The quantity and quality of each of the pools was assessed by Bioanalyser and subsequently by qPCR.

**Sequencing and mapping**. Libraries were sequenced on an Illumina HiSeq 2000. Reads were mapped to the *D. pseudoobscura* genome v3.1, and indexed using bowtie2[46]. Paired-end reads were aligned with option '-g 1 –library-type fr-secondstrand' using TopHat2.0.8b (which calls bowtie2.1.0[47]). The option '-g 1' instructs TopHat2 to report the best alignment to the reference for a given read. Exon features were counted using HTSeq-count[48] and the reads of all exons of each gene were combined to provide overall measures of gene expression.

**Statistical analysis**. The Baseline libraries (which were technically replicated four times) were used to define sex-biased genes, (separately for the four combinations of tissue and courtship status) and genes responding to courtship (separately for the four combinations of sex and tissue) independently of the mating system treatment groups. Sex-biased genes were defined with a FDR threshold of 5% and a requirement for absolute logFC of at least one, to minimise the allometric contributions to sex differences[38]. Courtship-biased and genes responding to mating system manipulation were defined at the 10% FDR threshold to allow the inclusion of more genes in analyses of DE genes. General patterns in the results were consistent with either 5 or 10% FDR.

The high and low sexual selection regimes were biologically replicated four times each at the establishment of the experimental evolution experiment, and one RNAseq library was obtained from each replicate, for each combination of tissue type, sex and courtship status. Preliminary analysis showed that expression data differed substantially between tissue type and sex, and also that sex bias was sensitive to courtship status. Since sex bias was the main focus of our study of the effects of mating system manipulation, we analysed separately all eight possible combinations of sex, courtship status and tissue, always including the technically replicated relevant baseline to minimise dispersion.

To detect differentially expressed genes, count data for each replicate were analysed with edgeR v3.18.1[18] running in R v.3.4.0[49]. Only genes with an average normalised count per million across all libraries used in each analysis >0 were included in analyses, and libraries were normalised with the default edgeR normalisation procedure. Dispersion was measured with default parameters using a negative binomial model. Genes with very low expression in one sex were retained as their exclusion did not qualitatively affect the results.

We performed GO enrichment analysis of DE genes based on the sexual selection response contrasts, or the virgin-courtship comparison within Baseline, using topGO v2.22.0 with the weight01 algorithm option to account for GO topology (https://bioconductor.org/packages/release/bioc/html/topGO.html). Results with *p* < 0.1 on Fisher's exact tests were retained (Supplementary Data 1 and 2). The background gene data set against which to test for enrichment was obtained separately for each contrast. Heatmaps were based on moderated log-counts-per million, obtained with the cpm function of edgeR and the average clustering method of hclust in R v3.4.0[49].

Following Hollis et al.[9], we tested mean expression differences between E and M treatments for all pairs of male-biased female-biased and unbiased genes in each of the eight sexual selection response contrasts using paired Mann–Whitney rank tests for each comparison, and performed a one-way ANOVA across gene classes. We repeated this after removing genes not expressed in one sex in our data, and after removing DE genes between testes and ovaries as identified in the tissue-specific *D. pseudoobscura* omnibus data set (GSE52058, https://www.ncbi.nlm.nih.gov/pubmed/25031657). In addition, we tested if the fold-change between treatments of each category of gene differed from zero using a bootstrapping approach resampled from the observed values with 10,000 replicates.

We present two further analyses based on randomisation of the direction of changes in gene expression with respect to sexual selection treatment. We call the first 'balanced randomisation', because it overcomes any inflation of variation that ordinary randomisation would produce by randomising equal numbers of samples from the two treatments. We derive a new metric, the 'standardised logFC value' (stdLogFC), by adjusting the *p*-value for each gene from the contrast between sexual selection treatments from the edgeR analysis. stdLogFC has the advantage that it considers both the magnitude of gene expression difference between treatments, and the confidence for the value obtained for each gene, because it takes into account the variance in expression of each gene across the samples. We converted these *p*-values into quantiles of a standard normal distribution to reduce the effect of outliers and account for their unknown distribution, and applied a sign to signify which treatment has the greater expression (E > M being positive). Our test statistic was obtained by comparing the average stdLogFC for each sex-biased gene set to that obtained by resampling (with 10,000 iterations) the stdLogFC of each gene set, after randomly inverting the stdLogFC sign for half of the genes. This inversion effectively randomises the influence of the sexual selection treatment.

We also used a more typical randomisation approach, by randomising individual replicates (libraries). This randomisation is theoretically superior to the 'balanced randomisation' across genes because it does not change any covariance structure possessed by the data set and accounts for outlier libraries that may disproportionately influence the result. However, it is worse in practice, because our data set was relatively small: randomisation of the treatment effect by constructing groups of four replicates, each containing two libraries from each treatment, resulted in only 36 possible such groups. Consequently, only a total of 630 (36 × 35/2) iterations could be conducted to represent the null distribution for our test statistic (observed mean stdLogFC), so $p$-values cannot be less than 1/630.

For each sexual selection response contrast, we tested if sex-biased genes differed in the likelihood of being DE using $\chi^2$ tests on the proportion of DE genes classified by their sex-bias status. We also performed $\chi^2$ tests on the proportion of DE genes in each of the *D. pseudoobscura* 5 Muller elements, on pools of all DE genes in each major comparison: courtship bias, sex bias and sexual selection response. The X chromosome arms were analysed both together and separately. Finally we performed a $\chi^2$ test on the proportion of the genes in the pool of genes DE by treatment which had homologues in *D. melanogaster* compared to that of all genes. All $\chi^2$ tests were conducted in R v4.3.0[49].

We downloaded the fly head normalised count data used by Hollis et al.[9] from the Gene Expression Omnibus accession number GSE50915 and estimated the average expression per gene for each treatment (high and low sexual selection). This was correlated with the average expression level of the homologues of these genes, from our data. We also correlated (Pearson correlation coefficients on logFC differences) the expression level of the same differentially expressed genes in males and females in our data, and generated 95% CIs from 10,000 bootstraps to compare these.

**Code availability**. Scripts related to the analysis presented here are archived at Zenodo.org with the identifier doi: 10.5281/zenodo.1044633[50].

**Data availability**. The data sets generated during the current study are available in the ArrayExpress repository with accession number E-MTAB-4486, https://www.ebi.ac.uk/arrayexpress/experiments/E-MTAB-4486/.

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

## Acknowledgements

The study was funded by a Natural Environment Research Council grant (NE/I014632/1) to M.G.R., R.R.S. and A.R.C., and an associated NBAF award (NBAF654). Brian Hollis, Tad Kawecki, Darren Parker and Elina Immonen provided many constructive discussions and advice. Rebecca Bastin, Margaret Hughes, Pia Koldkjaer and John Kenny all gave logistical support.

## Author contributions

M.G.R. and R.R.S. conceived the project, designed the experiments and supervised the work. M.G.R., R.R.S. and A.R.C. obtained funding for the work. Y.F. and A.R.C. advised with statistical and genomic analyses. P.V. completed the data collection and analysis. M.G.R., R.R.S. and P.V. wrote the paper, with input from all authors.

## Additional information

**Competing interests:** The authors declare no competing financial interests.

