## [Peer Review File · Nature Communications]

Reviewers' comments:

Reviewer #1 (Remarks to the Author):

Review of Veltsos et al

This is a very nice manuscript describing transcriptome divergence between lines of *Drosophila pseudoobscura* experimentally evolved with monogamy vs. polygandry. The results differ markedly from an earlier experiment in a related species (*D. melanogaster*) that also compared experimental populations evolved with or without enforced monogamy (Hollis et al 2014 Nat. Comm.). The earlier study found a very simple pattern, monogamy populations evolved increased expression of female biased genes and reduced expression of male biased genes. The patterns in the current study are quite different. Moreover, the authors have also collected data on expression of flies with courtship experience (unlike the previous work looking at naïve virgins) and found that evolved differences are not the same when examining virgins compared to non-virgins. All together, this study provides new and interesting results that over-turn the simplistic view the field was adapting, partly due to that Hollis et al. paper. If the results are statistically solid, this work is an important addition to the literature beyond Hollis et al. and earlier work from this group (Immonen et al. 2014). This manuscript should be published in Nature Communications eventually. I do have some points of major confusion that the authors must fix. My recommendation for publication assumes the author have reasonable answers to all of my issues or can fix these problems and retain the important claims.

My major concern is the lack of detail provided regarding statistical tests. Despite the Methods having a section laughably called "Statistical analysis", the tests underlying the major claims in the manuscript are not described there!

For example, the statistical tests alluded to in Fig. 1 need more explanation. What precisely is being compared "...directly (left)" as well as "...through comparisons with unbiased gene expression level (right)"?

I am unsure what analysis they have done but the Mann-Whitney analysis (referred to in Fig. 1) assumes that the genes are independent, which they are certainly not. The observations are independent at the level of the population (I believe they have 4 M and 4 E populations). A proper analysis (which people doing transcriptomics often forego) would use a summary measure for each population, correctly treating "population" as its unit of replication. At the moment, the analysis done here (and in Hollis et al), drastically overestimates the true number of degrees of freedom, artificially inflating power (possibly providing a simple explanation for the discrepancy between the studies). As it stands, a single weird replicate could drive a lot of these results (also true for Hollis et al.) but by considering each gene as an independent observation, you essentially treat environmental effects / measurement error as if it is a signal of among-treatment differences.

If they are going to analyze each gene as if it was an independent unit (which they shouldn't), why are they comparing sex-biased genes to unbiased genes? Why not just ask

if the mean expression difference between M and E for each gene category is more or less than 0 (assess by bootstrap CIs)?

The claim that "female-biased genes were upregulated in abdomens of virgin females under monogamy" seems hard to believe based on Fig. 1A (even when I blow up the figure to 400%!).

I don't understand the statement (Methods) that "Only genes with counts >10 were included in analyses..." Every important analysis involves a comparison between M and E and involves 4 replicates of each. Does that mean there had be more than 10 reads in all 8 samples. Please be clear.

Figs. 1 and 2. There appears to be no "unbiased" genes in the histogram. Is that correct? The figure is confusing as the key (inset into the histogram) implies they should be there.

Results: "Although there are only a few such genes, this allows us to ask if either type of sex-biased loci..." There are less than 8 genes in each case. Please state that explicitly, rather than making readers look in the SI.

Reviewer #2 (Remarks to the Author):

The authors use RNA-seq to compare gene expression in *D. pseudoobscura* flies maintained for over 150 generations with either enforced monogamy or elevated polyandry. They look at both abdomens and heads and also at virgin and courted flies. In particular, they focus on changes in sex-biased gene expression. Although they find significant changes in sex-biased gene expression associated with mating system, there is not a consistent pattern of masculinisation or feminisation: these patterns seem to vary among sexes and tissues.

The experiments and analyses presented in this paper appear to be solid and the authors do a good job of explaining their expectations and results. The major conclusion that comes from this work is that the previous observations made by Hollis et al. using a similar approach in *D. melanogaster* cannot be generalised, even to another *Drosophila* species. Indeed, the present study does not find a consistent pattern and concludes that "the direction of changes in expression of sex-biased genes is unpredictable". On its own, this finding is worthy of publication somewhere, but probably does not present the conceptual advance that the editors of Nature Communications have asked for. However, since the work directly contradicts a previous publication in Nature Communications, the editors may want to consider publishing it in the same journal.

In both this study and the study of Hollis et al., the changes in sex-biased expression induced by manipulating the mating system are relatively small. Indeed, only a small number represent statistically significant changes in sex-biased expression. It is mainly when all changes, regardless of individual significance, are analysed that significant changes

in the overall distribution can be observed. So, to me, it is not that surprising that the different studies get different results. A major contributor is likely to be the variation that is segregating in the initial population. This differs between studies, as the current study uses natural variation, while the Hollis et al. study used mutagenized flies. This may have implications for how much sexually antagonistic variation (and the type/extent of the antagonism) is present in the base population. There could be some variants that affect the expression of multiple genes. For example, there could be a single variant that increases the expression of dozens of male-biased genes. If such a variant happens to be in the base population and is selected (or drifts to high frequency) in one of the treatment populations, it would have a large effect on the outcome. Typically one assumes that each gene changes in expression independently, but this may not be the case. If it is not, then one would expect a higher variance in the results due to idiosyncrasies of the particular variants that are present in the population. This, again, could lead to differences between experiments or between tissues/sexes.

Minor comments:

1. Abstract: "experienced individuals have a more strongly masculinised transcriptome" - does "individuals" refers to both males and females? This is confusing, because the preceding sentence states that "courtship influences gene expression in males and females differently".
2. Page 4, last paragraph: I think the comma should be deleted in the first line. Also "in these tissues" would probably be better as "in this body segment". Finally, "the opposite pattern is seen in abdomens" - it would be clearer to write "in male abdomens".
3. Page 5: "the male-biased genes increase in expression more than that of female-biased genes" - this is not phrased correctly. Maybe it should be "genes" and "was more than"? Or changed to "increased" and delete "that"?
4. Page 5: "this allows us to ask if either type of sex-biased loci are more likely to respond to the manipulation as well as if they are more likely to be up- or down-regulated under monogamy". Here I think the subject is "type", so "are" and "they" don't agree.
5. Materials and Methods: it would be helpful to mention something about the population sizes of the base population and the monogamy/polyandry treatments. Is there a chance for deleterious alleles to drift to high frequency? Also, it would be helpful to mention the read length of the RNA-seq.
6. Figures 1 and 2: is it possible to reduce the scale of the X-axis so that one can see a better spread of the distribution? As it is, everything is grouped very tightly at the center, with only a few outliers. Maybe extreme outliers can be excluded from the figure? Or grouped together in a bin at the edges?

Reviewer #3 (Remarks to the Author):

Veltsos et al. have used experimental evolution to assess the effects of manipulating mating systems on sex-biased gene expression in *D. pseudoobscura*. Specifically, they subjected this normally polyandrous species to 150 generations of experimental monogamy and enhanced polyandry and then looked at changes in sex-biased gene expression (head and abdomen) relative to the baseline in the original parent population. A similar RNA-seq analysis was also conducted in flies that had commenced mating, but prior to the initiation of sperm/SFP transfer, to assess the influence of courtship on gene expression. The majority of the manuscript focuses on comparing the results of the current study to those reported by Hollis et al. (Nat. Comm 2014), which conducted a similar experiment in *D. melanogaster*. The overarching message of the manuscript is that the studies resulted in different, and oftentimes contradictory results. The fact that the studies generated different patterns of feminization and masculinization of gene expression has the potential to be informative in relation to sexual antagonism and the evolution of sex-biased genes expression. However, although similar in design, this study and that of Hollis differ in fundamental and important ways. Because of this, it isn't ultimately clear what is contributing to the differences and, more importantly, if the differences are truly reflective of the "dynamic and unpredictable" nature of sex-biased expression optima. A greater focus on (and much more in-depth analysis of) the data in hand, particularly the novel data relating to courtship, might prove to be more informative and would represent a more substantial advancement in the field. A key analysis to included would be one that delineates the contribution of male and female gonad expression to the masculinization and feminization of abdomen transcriptomes. It would seem that this is an important analysis to perform to take advantage of the focus on abdomen gene expression in the current study (rather than whole fly as in Hollis et al). Overall, the study is methodologically sound, both from the standpoint of RNA-seq experiments and statistical analyses. Although the genomic patterns of masculinization and feminization of expression are potentially interesting, a greater focus on the actual genes with altered expression would be welcome to ultimately understand if similar genes (between species, sexes and tissues) are subject to intragenomic conflict in relation to sex-specific expression optima and to the functions to which these gene contribute.

Major points:

1. Given the manuscript's emphasis on comparisons with the Hollis et al study, one would ideally hope to be able to distinguish differences that are associated with experimental design and those that can, with some confidence, be attributed to the unpredictable and dynamic nature of sex-biased gene expression evolution. The studies use different species (which probably have meaningful differences in their mating systems in the wild), different input populations with potential differences in standing genetic variation, different control expression datasets (the one used in the current study is superior, in my mind), different mating selection (polyandry vs. polygynandry), and examine different tissues, in the case of whole fly vs. abdomen. All of these differences have the potential to be informative to understanding the responses observed in this study but, as presented, discriminating between them in an informative manner seem unlikely. A reduced emphasis on comparing the studies and a greater focus on the novel observations in this experiment might be suggested . The following are some possible analyses that might assist in inferring the basis

of differences between the studies.

2. Abdomen vs. whole fly: It isn't clear why the current study doesn't establish the contribution of testis and ovary expressed genes to the patterns of abdomen masculinization and feminization. The necessary data are available (Gene Omnibus: GSE52058) and the previous study did a similar analysis (although they did not pursue this in much depth). Differences between whole fly and abdomen results could be explained by the greater influence of gonads in the later. It would be useful to know if changes in testis or ovary expression optima are similar or different between species (either in magnitude or composition). As orthology relationships are available between these species, it would be rather straightforward to look for correlations between the two studies in relation to shared gonad expressed genes (testis or ovary-biased or specific) and those not expressed in gonads. The relative contribution of gonads to sex-biased gene expression and masculinization/feminization, relative to somatic tissues, would be very interesting in and of itself. This suggests relates, in part, to the discussion of allometry in the discussion. While I agree with the author's statement about allometry in relation to head and abdomen patterns, various analyses could be done to support this statement.

3. Head: Here a direct comparison of results is possible and it would be useful to establish (1) the extent to which sex-biased expression is shared or divergent between species (e.g. do the same genes contribute in both species and in the same direction), (2) whether masculinization and feminization in the two studies, respectively, are associated with a core set of genes (or GO functions) and (3) is there a greater overlap in changes in gene expression between head and abdomen than might be expected and do these show correlated responses in *D. mel.* There is also the possibility that differences between species are due to genes which do not share orthology between species. In other words, do paralogs or lineage specific genes contribute disproportionately to the differential shift in overall expression.

4. Abdomen and head analysis: What is the genomic distribution of genes experiencing masculinized or feminized expression and how does this relate to the overall distribution of sex-biased gene expression? How does the X contribute to this (in males and females) and does this underly differences between species?

5. This analysis, as well as Hollis et al., looks at the change in gene expression in males and females independently. However, the hypothesis of sexual antagonism predicts that relaxed sexual selection under monogamy will cause movement of gene expression toward the female optimum in both sexes. Examining correlations between male and female changes in gene expression can test this prediction. This additional analysis will also potentially offer a new way to categorize genes expression changes in males and females as either correlated or divergent and examine potential differences between these categories and between species.

6. The authors highlight the identification of 7 significantly differentially expressed genes following experimental evolution. However valuable information about the identity and gene ontology of these genes is absent from the manuscript. The GO categories represented by

these genes could be discussed both in the results and the discussion. Although Fig. S1 is consistent with the display of data in the other figures it may not be the most informative way to show this data. A heat map that shows clustering, a visual representation of expression levels, and includes gene names would be much more informative. Similarly, the discussion of significant gene expression changes following courtship should also be discussed in the results prior to the paragraph in the discussion section. Fig. S2 may also be more informative as a heat map or a different format that includes gene names. (There was an error in the format of table S1 and it did not appear in my copy of the document document, it is possible that this table, with modification to include the 7 genes differentially expressed in virgins following experimental evolution treatments, could address the comments above)

7. The gene ontology analysis could be even more informative if expanded beyond exclusively differentially expressed genes. In particular, although only 7 genes were differentially expressed following experimental evolution mating regimes, it would be valuable to delve into GO patterns in genes that show trends in expression changes over a certain cutoff. Expanding this analysis would also provide an opportunity to directly compare to the GO of genes with non-significant changes in expression in *D. melanogaster*.

Minor comments

1. Although the authors clearly explain the differences in the set up of polyandry vs. polygynandrous mating systems in this study vs. Hollis et al., in the discussion, this information would be beneficial to mention earlier. For example in the first paragraph of the results section the authors contrast their dissection techniques to the Hollis et al. study, but do not mention the differences in artificial evolution between the studies.

2. The gene ontology data is especially interesting, and an excellent additional analysis that was missing from the Hollis et al. study. However, it is limited to the discussion and would be informative as part of the main results.

3. In the discussion of antagonistic selection and costs to female in polygamy (2nd paragraph of discussion "for example, different levels of male harm..." It would be valuable to also cite/discuss Gerrard et al. 2013 Genome-Wide Responses of Female Fruit Flies Subjected to Divergent Mating Regimes on gene expression changes that might be involved in decreased female longevity under high mating regimes.

4. To parse apart the effects of courtship vs. intromission vs. mating vs. sperm/SFPs more fully it would be interesting to also cite/discuss McGraw et al. 2004 Genes Regulated by Mating, Sperm or Seminal Fluid Proteins in Mated Female *Drosophila melanogaster* and the comparison of the genes they identify as being regulated by non-sperm/non-Acp components of mating to the genes this study found differentially regulated by courtship.

5. The description of statistical methods is not sufficient to easily understand the analysis performed. The figure legends describe Mann-Whitney rank tests to compare changes in sex biased gene expression that are not mentioned in the methods. The methods section also

lacks any mention of the χ^2 test used to compare proportions of differentially expressed genes in the head and abdomen following mating.

6. The large range included in the X-axis of the figures makes it very difficult to visualize the graphs and the changes that are predominantly occurring between -5 and 5 log change in expression. Figure legends could be more informative in describing the significant changes in gene expression observed in the panels.

7. p4; 3rd para: "...In contrast the opposite pattern is seen in abdomens, where female biased-genes were..." The previous sentence discussed abdomens of virgin females and this sentence appears to refer to males but "male" is missing.

8. p7; 1st para; line 5: unpaired parentheses

9. p7; 2nd para; 1st sentence: "is" to "are"

10. p7; 3rd para: unformatted citation

Reviewers' comments:

Reviewer #1 (Remarks to the Author):

Review of Veltsos et al

This is a very nice manuscript describing transcriptome divergence between lines of *Drosophila pseudoobscura* experimentally evolved with monogamy vs. polygandry. The results differ markedly from an earlier experiment in a related species (*D. melanogaster*) that also compared experimental populations evolved with or without enforced monogamy (Hollis et al 2014 Nat. Comm.). The earlier study found a very simple pattern, monogamy populations evolved increased expression of female biased genes and reduced expression of male biased genes. The patterns in the current study are quite different. Moreover, the authors have also collected data on expression of flies with courtship experience (unlike the previous work looking at naïve virgins) and found that evolved differences are not the same when examining virgins compared to non-virgins. All together, this study provides new and interesting results that over-turn the simplistic view the field was adapting, partly due to that Hollis et al. paper. If the results are statistically solid, this work is an important addition to the literature beyond Hollis et al. and earlier work from this group (Immonen et al. 2014). This manuscript should be published in Nature Communications eventually. I do have some points of major confusion that the authors must fix. My recommendation for publication assumes the author have reasonable answers to all of my issues or can fix these problems and retain the important claims.

My major concern is the lack of detail provided regarding statistical tests. Despite the Methods having a section laughably called "Statistical analysis", the tests underlying the major claims in the manuscript are not described there!

For example, the statistical tests alluded to in Fig. 1 need more explanation. What precisely is being compared "...directly (left)" as well as "...through comparisons with unbiased gene expression level (right)"?

We apologize that the reviewer found this section lacking in details. We have strived to ensure that the revision carefully and completely describes the statistical tests used for each component of the work in our revised "Statistical analysis" section (starting line 440) and throughout the text.

I am unsure what analysis they have done but the Mann-Whitney analysis (referred to in Fig. 1) assumes that the genes are independent, which they are certainly not. The observations are independent at the level of the population (I believe they have 4 M and 4 E populations). A proper analysis (which people doing transcriptomics often forego) would use a summary measure for each population, correctly treating "population" as its unit of replication. At the moment, the analysis done here (and in Hollis et al), drastically overestimates the true number of degrees of freedom, artificially inflating power (possibly providing a simple explanation for the discrepancy between the studies). As it stands, a single weird replicate could drive a lot of these results (also true for Hollis et al.) but by considering each gene as an independent observation, you essentially treat environmental effects / measurement error as if it is a signal of among-treatment differences.

Because one of the motivations of this manuscript was to examine if we obtain a similar result to Hollis et al, repeating their analytical methodologies seemed appropriate, so we leave the Mann-

Whitney tests in this presentation. However, we do appreciate the point that gene expression changes may not be independent. We have examined this issue further. We add heatmaps for DE genes (in response to a comment from Reviewer 3 about replicate consistency) which show that for virtually all analyses the replicate lines cluster together, along with a higher profile to our functional analysis of the most responsive genes, which examines the functions represented rather than treats each gene independently. Note also that essentially the same patterns occur when we simply do chi squared tests on simple counts of the most divergent genes (with either 5 or 10% FDR cut offs) so the main results are robust.

If they are going to analyze each gene as if it was an independent unit (which they shouldn't), why are they comparing sex-biased genes to unbiased genes? Why not just ask if the mean expression difference between M and E for each gene category is more or less than 0 (assess by bootstrap CIs)?

We are surprised by this comment. The argument arising from comparative genomic studies in the literature, and in Hollis et al., is that sex-biased genes respond to mating system changes more than un-biased genes. Testing this requires comparing the responses between these types of genes. However, we appreciate the point that we could also test responses of each class of gene separately. We have now done this by bootstrapping and every comparison confirms our conclusions based on the Mann-Whitney tests. This is added as a supplementary table (Table S1).

The claim that "female-biased genes were upregulated in abdomens of virgin females under monogamy" seems hard to believe based on Fig. 1A (even when I blow up the figure to 400%!).

This is clearer in the new expanded figure, and is also confirmed by the new bootstrapping analysis (Table S1).

I don't understand the statement (Methods) that "Only genes with counts >10 were included in analyses..." Every important analysis involves a comparison between M and E and involves 4 replicates of each. Does that mean there had be more than 10 reads in all 8 samples. Please be clear.

This is now clarified (line 458). There had to be more than 10 reads in total for the 8 samples.

Figs. 1 and 2. There appears to be no "unbiased" genes in the histogram. Is that correct? The figure is confusing as the key (inset into the histogram) implies they should be there.

There are no unbiased genes in the histogram but they are included in the 'stem and leaf' plot below to allow visual comparison with the changes in sex-biased genes (as justified above). The legend clarifies this.

Results: "Although there are only a few such genes, this allows us to ask if either type of sex-biased loci..." There are less than 8 genes in each case. Please state that explicitly, rather than making readers look in the SI.

We have omitted this section and present a new table (Table 1) including gene number and chi

squared tests based on FDR 10%, as requested by reviewer 3.

Reviewer #2 (Remarks to the Author):

The authors use RNA-seq to compare gene expression in *D. pseudoobscura* flies maintained for over 150 generations with either enforced monogamy or elevated polyandry. They look at both abdomens and heads and also at virgin and courted flies. In particular, they focus on changes in sex-biased gene expression. Although they find significant changes in sex-biased gene expression associated with mating system, there is not a consistent pattern of masculinisation or feminisation: these patterns seem to vary among sexes and tissues.

The experiments and analyses presented in this paper appear to be solid and the authors do a good job of explaining their expectations and results. The major conclusion that comes from this work is that the previous observations made by Hollis et al. using a similar approach in *D. melanogaster* cannot be generalised, even to another *Drosophila* species. Indeed, the present study does not find a consistent pattern and concludes that "the direction of changes in expression of sex-biased genes is unpredictable". On its own, this finding is worthy of publication somewhere, but probably does not present the conceptual advance that the editors of Nature Communications have asked for. However, since the work directly contradicts a previous publication in Nature Communications, the editors may want to consider publishing it in the same journal.

In both this study and the study of Hollis et al., the changes in sex-biased expression induced by manipulating the mating system are relatively small. Indeed, only a small number represent statistically significant changes in sex-biased expression. It is mainly when all changes, regardless of individual significance, are analysed that significant changes in the overall distribution can be observed. So, to me, it is not that surprising that the different studies get different results. A major contributor is likely to be the variation that is segregating in the initial population. This differs between studies, as the current study uses natural variation, while the Hollis et al. study used mutagenized flies. This may have implications for how much sexually antagonistic variation (and the type/extent of the antagonism) is present in the base population. There could be some variants that affect the expression of multiple genes. For example, there could be a single variant that increases the expression of dozens of male-biased genes. If such a variant happens to be in the base population and is selected (or drifts to high frequency) in one of the treatment populations, it would have a large effect on the outcome. Typically one assumes that each gene changes in expression independently, but this may not be the case. If it is not, then one would expect a higher variance in the results due to idiosyncrasies of the particular variants that are present in the population. This, again, could lead to differences between experiments or between tissues/sexes.

These are very interesting comments which we appreciate and largely agree with, but are hard to address directly. As stated in response to reviewer #1, we have added heatmaps, better functional analyses and bootstrap analyses to confirm the robustness of our results, and we find the same direction of change when limited to only DE genes. A new graphical summary (Figure S4) indicates out robust major results clearly.

Minor comments:

1. Abstract: "experienced individuals have a more strongly masculinised transcriptome" - does "individuals" refer to both males and females? This is confusing, because the preceding sentence states that "courtship influences gene expression in males and females differently".

We have amended this.

2. Page 4, last paragraph: I think the comma should be deleted in the first line. Also "in these tissues" would probably be better as "in this body segment". Finally, "the opposite pattern is seen in abdomens" - it would be clearer to write "in male abdomens".

All amended.

3. Page 5: "the male-biased genes increase in expression more than that of female-biased genes" - this is not phrased correctly. Maybe it should be "genes" and "was more than"? Or changed to "increased" and delete "that"?

Clarified

4. Page 5: "this allows us to ask if either type of sex-biased loci are more likely to respond to the manipulation as well as if they are more likely to be up- or down-regulated under monogamy". Here I think the subject is "type", so "are" and "they" don't agree.

Fixed

5. Materials and Methods: it would be helpful to mention something about the population sizes of the base population and the monogamy/polyandry treatments. Is there a chance for deleterious alleles to drift to high frequency? Also, it would be helpful to mention the read length of the RNA-seq.

It is important to note that any such drift effects should only contribute noise to the results, not significant differences between replicated treatments. As in Snook et al. 2009, the N_e of all M and E lines are ca. 120, originally set up from the baseline population. As the baseline population is kept en masse, then N_e will be greater than 120 but has not been estimated. This information has been added to Materials and Methods, starting line 405 in section "Fly maintenance and preparation".

6. Figures 1 and 2: is it possible to reduce the scale of the X-axis so that one can see a better spread of the distribution? As it is, everything is grouped very tightly at the center, with only a few outliers. Maybe extreme outliers can be excluded from the figure? Or grouped together in a bin at the edges?

We have amended the range of the X axis as requested and produced revised figures.

Reviewer #3 (Remarks to the Author):

Veltsos et al. have used experimental evolution to assess the effects of manipulating mating systems on sex-biased gene expression in *D. pseudoobscura*. Specifically, they subjected this normally polyandrous species to 150 generations of experimental monogamy and enhanced polyandry and then looked at changes in sex-biased gene expression (head and abdomen) relative to the baseline in the original parent population. A similar RNA-seq analysis was also conducted in flies that had commenced mating, but prior to the initiation of sperm/SFP transfer, to assess the influence of courtship on gene expression. The majority of the manuscript focuses on comparing the results of the current study to those reported by Hollis et al. (Nat. Comm 2014), which conducted a similar experiment in *D. melanogaster*. The overarching message of the manuscript is that the studies resulted in different, and oftentimes contradictory results. The fact that the studies generated different patterns of feminization and masculinization of gene expression has the potential to be informative in relation to sexual antagonism and the evolution of sex-biased genes expression. However, although similar in design, this study and that of Hollis differ in fundamental and important ways. Because of this, it isn't ultimately clear what is contributing to the differences and, more importantly, if the differences are truly reflective of the "dynamic and unpredictable" nature of sex-biased expression optima. A greater focus on (and much more in-depth analysis of) the data in hand, particularly the novel data relating to courtship, might prove to be more informative and would represent a more substantial advancement in the field. A key analysis to included would be one that delineates the contribution of male and female gonad expression to the masculinization and feminization of abdomen transcriptomes. It would seem that this is an important analysis to perform to take advantage of the focus on abdomen gene expression in the current study (rather than whole fly as in Hollis et al). Overall, the study is methodologically sound, both from the standpoint of RNA-seq experiments and statistical analyses. Although the genomic patterns of masculinization and femization of expression are potentially interesting, a greater focus on the actual genes with altered expression would be welcome to ultimately understand if similar genes (between species, sexes and tissues) are subject to intragenomic conflict in relation to sex-specific expression optima and to the functions to which these gene contribute.

We appreciate these points, and have emphasised the significance of our novel results more with additional analyses and discussion. Especially, we have included the heatmap, and a new GO term analysis of the most divergent loci between the lines and tissues, as well as an expanded analysis of the loci responding to selection. This does highlight some interesting gene sets. We have also examined the effect of removing genes expressed in gonad tissue (an excellent suggestion) and compared our results more quantitatively with Hollis et al in places.

Major points:

1. Given the manuscript's emphasis on comparisons with the Hollis et al study, one would ideally hope to be able to distinguish differences that are associated with experimental design and those that can, with some confidence, be attributed to the unpredictable and dynamic nature of sex-biased gene expression evolution. The studies use different species (which probably have meaningful differences in their mating systems in the wild), different input populations with potential differences in standing genetic variation, different control expression datasets (the one used in the current study is superior, in my mind), different mating selection (polyandry vs. polygynandry), and examine different tissues, in the case of whole fly vs. abdomen. All of these differences have the potential to be informative to understanding the responses observed in this

study but, as presented, discriminating between them in an informative manner seem unlikely. A reduced emphasis on comparing the studies and a greater focus on the novel observations in this experiment might be suggested. The following are some possible analyses that might assist in inferring the basis of differences between the studies.

Our experiment was never intended to be an exact replicate of Hollis et al., both arose independently (this experimental evolution study was started in 2002). It is impossible to say with any confidence which of the differences between the studies, including study species and social condition, may be responsible for the different results. We simply cannot reliably conclude which are due to the vagaries of sexual antagonism versus design or species differences. We try to emphasise this important issue more clearly in the revision (starting line 280).

2. Abdomen vs. whole fly: It isn't clear why the current study doesn't establish the contribution of testis and ovary expressed genes to the patterns of abdomen masculinization and feminization. The necessary data are available (Gene Omnibus: GSE52058) and the previous study did a similar analysis (although they did not pursue this in much depth). Differences between whole fly and abdomen results could be explained by the greater influence of gonads in the later. It would be useful to know if changes in testis or ovary expression optima are similar or different between species (either in magnitude or composition). As orthology relationships are available between these species, it would be rather straightforward to look for correlations between the two studies in relation to shared gonad expressed genes (testis or ovary-biased or specific) and those not expressed in gonads. The relative contribution of gonads to sex-biased gene expression and masculinization/feminization, relative to somatic tissues, would be very interesting in and of itself. This suggests relates, in part, to the discussion of allometry in the discussion. While I agree with the author's statement about allometry in relation to head and abdomen patterns, various analyses could be done to support this statement.

Thanks very much for these interesting suggestions. We have repeated our analyses taking advantage of the data quoted to examine the influence of gonad biased genes (section starting line 125). We find consistent patterns omitting them, and we discuss that they are not driving the patterns. We have correlated responses of head material between the two studies, but not gonad. There is no gonad-specific data in either our study or Hollis et al, and, as stated before, the platforms differ. These issues are now discussed, along with other factors adding noise in the comparison between the two studies, such as the lack of DE genes in Hollis et al, and suggest that future tissue specific studies are necessary for more conclusive results.

3. Head: Here a direct comparison of results is possible and it would be useful to establish (1) the extent to which sex-biased expression is shared or divergent between species (e.g. do the same genes contribute in both species and in the same direction), (2) whether masculinization and feminization in the two studies, respectively, are associated with a core set of genes (or GO functions) and (3) is there a greater overlap in changes in gene expression between head and abdomen than might be expected and do these show correlated responses in *D. mel.* There is also the possibility that differences between species are due to genes which do not share orthology between species. In other words, do paralogs or lineage specific genes contribute disproportionately to the differential shift in overall expression.

Again, while we completely understand why the reviewer thinks these analyses comparing the two studies are interesting, we are very cautious about how meaningful comparisons are in practise,

because of the large differences in methodologies and the noisiness and non-significance of the data in Hollis et al. We think that it is probably pushing their data too far.

Nevertheless:

- (1) We calculate the correlation in FC differences between monogamy and elevated sexual selection treatments in virgin heads between the two studies because these are comparable. Changes in the 10% DE genes from our study do not correlate with the homologues in *D. melanogaster*. These plots are added to the supplementary material (Figure S5 and section starting line 198).
- (2) We present more in depth GO term analysis of our results, including a new analysis of the differences between the lines. The lack of any DE in Hollis precludes any meaningful comparison with theirs.
- (3) There was virtually no overlap in DE genes between head and abdominal tissue in any of the comparisons (the greatest overlap included only 7 genes in common, in courted males, and this represents <3% of the DE genes). We have not presented this analysis.
- (4) Orthology. Our DE genes show high homology with genes in *D. melanogaster*, in fact against some expectations, our DE genes are more likely to have homologues, so there is no evidence of a disproportionate effect of lineage specific genes (this is now presented in the results, section “Correlations between sexes and species” starting line 198).

4. Abdomen and head analysis: What is the genomic distribution of genes experiencing masculinized or feminized expression and how does this relate to the overall distribution of sex-biased gene expression? How does the X contribute to this (in males and females) and does this underly differences between species?

We have added a new analysis of the chromosomal distribution of genes showing DE in our analysis, see section “Chromosomal distribution of DE genes” (starting line 217).

5. This analysis, as well as Hollis et al., looks at the change in gene expression in males and females independently. However, the hypothesis of sexual antagonism predicts that relaxed sexual selection under monogamy will cause movement of gene expression toward the female optimum in both sexes. Examining correlations between male and female changes in gene expression can test this prediction. This additional analysis will also potentially offer a new way to categorize genes expression changes in males and females as either correlated or divergent and examine potential differences between these categories and between species.

We have added a correlation analysis of the changes in expression for the DE genes between the sexes (see section “Correlations between sexes and species” starting line 198). This shows that responses are correlated, perhaps suggesting constraints on sex-specific responses. However, there is no evidence that the strength of these correlations differ between the treatments.

6. The authors highlight the identification of 7 significantly differentially expressed genes following experimental evolution. However valuable information about the identity and gene ontology of these genes is absent from the manuscript. The GO categories represented by these genes could be discussed both in the results and the discussion. Although Fig. S1 is consistent with the display of data in the other figures it may not be the most informative way to show this data. A heat map that shows clustering, a visual representation of expression levels, and includes gene names would be

much more informative. Similarly, the discussion of significant gene expression changes following courtship should also be discussed in the results prior to the paragraph in the discussion section. Fig. S2 may also be more informative as a heat map or a different format that includes gene names. (There was an error in the format of table S1 and it did not appear in my copy of the document document, it is possible that this table, with modification to include the 7 genes differentially expressed in virgins following experimental evolution treatments, could address the comments above)

We have added heatmaps and an analysis of GO terms in the DE genes, and supplementary tables of these GO terms.

7. The gene ontology analysis could be even more informative if expanded beyond exclusively differentially expressed genes. In particular, although only 7 genes were differentially expressed following experimental evolution mating regimes, it would be valuable to delve into GO patterns in genes that show trends in expression changes over a certain cutoff. Expanding this analysis would also provide an opportunity to directly compare to the GO of genes with non-significant changes in expression in *D. melanogaster*.

We changed our FDR to 10% throughout, which allows better functional analysis. We state that similar patterns are detected when restricted to a more conservative 5%. We think these improved GO analyses improve the study substantially. However, as stated before, we do not choose to present comparisons with the *D. melanogaster* data, which did not do this and had no significant DE at the 5% level.

Minor comments

1. Although the authors clearly explain the differences in the set up of polyandry vs. polygynandrous mating systems in this study vs. Hollis et al., in the discussion, this information would be beneficial to mention earlier. For example in the first paragraph of the results section the authors contrast their dissection techniques to the Hollis et al. study, but do not mention the differences in artificial evolution between the studies.

With respect, we find embarking on a detailed description of the Hollis design immediately at the start of the results section to be quite distracting, and prefer to lead with our own results. We flag up that there are differences throughout the manuscript and discuss them in the discussion.

2. The gene ontology data is especially interesting, and an excellent additional analysis that was missing from the Hollis et al. study. However, it is limited to the discussion and would be informative as part of the main results.

This is added (section starting line 172).

3. In the discussion of antagonistic selection and costs to female in polygamy (2nd paragraph of discussion "for example, different levels of male harm...") It would be valuable to also cite/discuss Gerrard et al. 2013 Genome-Wide Responses of Female Fruit Flies Subjected to Divergent Mating

Regimes on gene expression changes that might be involved in decreased female longevity under high mating regimes.

We have added this reference (line 267).

4. To parse apart the effects of courtship vs. intromission vs. mating vs. sperm/SFPs more fully it would be interesting to also cite/discuss McGraw et al. 2004 Genes Regulated by Mating, Sperm or Seminal Fluid Proteins in Mated Female *Drosophila melanogaster* and the comparison of the genes they identify as being regulated by non-sperm/non-Acp components of mating to the genes this study found differentially regulated by courtship.

We have analysed these genes and we find no significant overlap in the loci detected. This is added to the discussion of genes changing expression due to courtship, line 191

5. The description of statistical methods is not sufficient to easily understand the analysis performed. The figure legends describe Mann-Whitney rank tests to compare changes in sex biased gene expression that are not mentioned in the methods. The methods section also lacks any mention of the χ^2 test used to compare proportions of differentially expressed genes in the head and abdomen following mating.

Reviewer 1 also drew attention to the statistical analysis section. As outlined in our response to reviewer 1, we have strived to ensure a more thorough explanation throughout and with regard to the χ^2 tests.

6. The large range included in the X-axis of the figures makes it very difficult to visualize the graphs and the changes that are predominantly occurring between -5 and 5 log change in expression. Figure legends could be more informative in describing the significant changes in gene expression observed in the panels.

We have amended the range of the X axis as requested and produced revised figures.

7. p4; 3rd para: "...In contrast the opposite pattern is seen in abdomens, where female biased-genes were..." The previous sentence discussed abdomens of virgin females and this sentence appears to refer to males but "male" is missing.

Fixed

8. p7; 1st para; line 5: unpaired parentheses

Fixed

9. p7; 2nd para; 1st sentence: "is" to "are"

Fixed

10. p7; 3rd para: unformatted citation

Fixed

Reviewers' comments:

Reviewer #1 (Remarks to the Author):

This could be a reasonably important paper; I respectfully disagree with Reviewer 2 of the original version. The Hollis et al paper paints such a pleasingly simple picture that it has influenced the field. The current work shows that a fairly similar experiment with a closely related species gives a substantially different result. If the Hollis et al result was robust and generalizable, the current authors should have seen something similar. Certainly, there are a number of plausible reasons for the difference between studies that the other reviewers mentioned in the last round (e.g., different species, different starting variation, differences in details of how treatments are carried out) and obviously things would be better if we knew why the results are so different. Nonetheless, I think the difference from the Hollis et al study alone makes it worthwhile. In addition, the contrast in expression between virgin and mated flies from different treatments adds a new and valuable piece.

I remain disappointed that the authors have continued to do an analysis that they should (and probably do know) is not really statistically valid. The authors acknowledged this concern in their review but did not really do much about it. The primary analyses treat genes as the independent unit of replication. This is wrong (note that the authors did not attempt to argue otherwise). The primary unit of analysis should be population. The key inferences they want to make involve contrasts between treatments (M versus E) for which they have 4 independent replicates of each (not hundreds or thousands as their gene-based analyses assume). I realize that the analysis they use is what Hollis et al did (and also what many other expression studies do) but evolutionary biologists should know better; sinking to the lowest common denominator does not seem like a good argument for using an analysis with the wrong number of degrees of freedom. The resulting false inflation of power could be a reason for false signals in either the current study or Hollis et al or both. I would be supportive of keeping the current analysis for comparative purposes but I strongly advocate doing a statistically valid analysis as the center piece. Creating a summary statistic for each population for each gene category (male-biased genes, female-biased, unbiased) would be the simplest way to go in order to do proper hypothesis testing but you'd have (appropriately) much less power because you'd only have $n = 4$ for each treatment. The heat maps are encouraging but if those are good enough then show those in the main text and leave off with p-values (as the ones presented aren't really legitimate).

In re-reading the Methods, I became worried that there might be unintended differences between virgin and non-virgin flies (Ln 413-416). I would like a clear and explicit statement about all the differences between virgin and non-virgin flies including age, handling, or any other differences they can think of.

Minor comments:

Ln 108: What does "marginally" mean here. This is a bit of a loaded word often meant to imply $0.05 < p < 0.1$.

Fig. 1&2. I could not find where the stars are explained in terms of specific p-value ranges.

Ln 178. Please provide a brief explanation of how/why glutathione metabolism is linked to sexual selection. I have no idea and I suspect most readers won't.

Reviewer #2 (Remarks to the Author):

The authors have addressed my previous concerns adequately and improved the manuscript. I have no further comments.

Reviewer #3 (Remarks to the Author):

Veltsos et al. have done a thorough job of addressing comments from all of the reviewers. In particular, the bootstrapping analysis of gene expression changes is an important statistical improvement. The integration of existing testis and ovary data, as well as the genomic distribution analysis (X vs. autosome), makes this study far more nuanced than the original Hollis et al study. The use of a baseline population for comparisons is a particular strength of the current study, as well as the choice of tissues and the novel data relating to post-copulatory expression. The depth of the analysis is now impressive and the complexity of the outcomes suggests that sexual antagonism and the evolution of sex-biased gene expression are perhaps far more complicated than many were beginning to assume. This is perhaps the most important contribution of the study and will hopefully motivate additional investigation. The importance of understanding genetic variation in the source population is discussed appropriately and is certainly something that is critical for future studies to take into consideration. I have a couple of additional suggestions that may be worth consideration prior to publication.

1. The changes in expression patterns are complex in nature and therefore somewhat difficult to keep straight when reading the manuscript. Might it be helpful to readers to move Fig S4 to the main text? This figure proved useful to me and is effective in summarizing the main trends.

2. I appreciate the inclusion of the new testis and ovary analyses and it is interesting to see that the patterns remain after removal of male:female gonad biased genes. In my initial comment, I was thinking that this data could be used to estimate the extent to which non-gonadal sex biased genes contribute to the overall patterns observed. As would be expected, there are far more genes that are sex-biased in the abdomen dataset relative to the brain but we don't know the full extent to which this is accounted for by "reproductive" genes. It is quite possible that other tissues in the abdomen (including both other reproductive and non-reproductive tissues) might contribute meaningfully to sex biased genes expression and gene expression evolution. Does this make sense? It would be interesting to estimate this and determine what non-gonadal tissues might be responsive to shifts in sexual selection. Instead of comparing testis to ovary to exclude genes (as you have done), I was thinking of comparing testis and ovary to whole male/female expression

to estimate gonad specificity and then determine if and to what extent somatic sex-biased genes respond to shifts in mating system. Sorry, I'm not sure I'm explaining this well.

3. I'm curious about why the "correlation between sexes" analysis was restricted to only significantly differentially expressed genes. The pattern emerging from this study (and the Hollis study) is small shifts in sex-biased expression optima across many genes and it would therefore be more informative to look for relationships across the full set of male and female biased genes. Perhaps a correlational analysis is not ideal in this case. One could simply ask if the directionality of change in males and females overlaps more than would be expected by chance. This would be relatively easy to test by simulation.

4. Similarly, I'm concerned the genomic distribution analysis may suffer from the same sort of problem. Unique patterns of change for sex-linked genes may be missed when focusing only on DE genes. Would it be possible to statistically compare the full distribution of gene expression changes between autosomes, XL and XR genes? Theory predicts differences in response between these classes of genes and it would be unfortunate to miss potentially informative patterns because the analysis focusses solely on FDR 10%.

Reviewers' comments:

Reviewer #1 (Remarks to the Author):

This could be a reasonably important paper; I respectfully disagree with Reviewer 2 of the original version. The Hollis et al paper paints such a pleasingly simple picture that it has influenced the field. The current work shows that a fairly similar experiment with a closely related species gives a substantially different result. If the Hollis et al result was robust and generalizable, the current authors should have seen something similar. Certainly, there are a number of plausible reasons for the difference between studies that the other reviewers mentioned in the last round (e.g., different species, different starting variation, differences in details of how treatments are carried out) and obviously things would be better if we knew why the results are so different. Nonetheless, I think the difference from the Hollis et al study alone makes it worthwhile. In addition, the contrast in expression between virgin and mated flies from different treatments adds a new and valuable piece.

I remain disappointed that the authors have continued to do an analysis that they should (and probably do know) is not really statistically valid. The authors acknowledged this concern in their review but did not really do much about it. The primary analyses treat genes as the independent unit of replication. This is wrong (note that the authors did not attempt to argue otherwise). The primary unit of analysis should be population. The key inferences they want to make involve contrasts between treatments (M versus E) for which they have 4 independent replicates of each (not hundreds or thousands as their gene-based analyses assume). I realize that the analysis they use is what Hollis et al did (and also what many other expression studies do) but evolutionary biologists should know better; sinking to the lowest common denominator does not seem like a good argument for using an analysis with the wrong number of degrees of freedom. The resulting false inflation of power could be a reason for false signals in either the current study or Hollis et al or both. I would be supportive of keeping the current analysis for comparative purposes but I strongly advocate doing a statistically valid analysis as the center piece. Creating a summary statistic for each population for each gene category (male-biased genes, female-biased, unbiased) would be the simplest way to go in order to do proper hypothesis testing but you'd have (appropriately) much less power because you'd only have $n = 4$ for each treatment. The heat maps are encouraging but if those are good enough then show those in the main text and leave off with p-values (as the ones presented aren't really legitimate).

We acknowledge that criticisms based on pooled estimates of dispersion applied at the gene level are valid, especially with respect to the degrees of freedom used in statistical testing (though this approach has been modelled at some length and is widely adopted - one of the key references behind it [Robinson et al. 2010 Bioinformatics 26:139-140] has been cited >4,000 times in Web of Science).

However, we do have issues with the alternative proposed by the reviewer. We believe that reducing the expression level of thousands of genes in one contrast to a single mean and testing for differences between 2 sets of 4 means with a t-test or similar statistic is less appropriate. The credibility of a test based on two means of 4 data values sets would be questionable (the normal distribution may be approximately followed by the two means based on central limit theorem, but in our particular case the approximation is unlikely to be good, given that each sample size is only 4). Moreover, information contained within the expression data for the individual genes would be lost. For example, it would be impossible to know whether any difference was contributed by several genes in the gene set or only by a few

outlier genes with an extreme expression difference. It is doing ‘statistics on statistics’ and we would argue strongly that doing so is in fact much more likely to produce biases than approaches which take into account the variance underlying these means (e.g. Morrissey, 2017 J. Evol. Biol. 29:1882-1904).

Instead, we think that we have derived a different approach which explicitly addresses the N=4 issue as requested, incorporates the observed distribution of gene counts, but makes no explicit assumptions about this distribution. For each type of gene (e.g. male-biased) we produced a set of read counts at the level of the replicate and calculated the difference in the 4 means, as suggested by the reviewer. We then resampled half the genes, randomised between treatments, and recalculated a mean between these randomised columns generating a null distribution of the difference between these 2 sets of four means if there were no difference due to treatment. We compared our observed difference with these null distributions. For virtually all contrasts the observed differences for the sex-biased genes were significant outliers, and the direction of change virtually always matches our original interpretations (one single exception is a case where there were relatively few sex-biased genes).

This is presented in detail as an additional supplement. We think this is more appropriate than making it the ‘centrepiece’ of the paper because it is unconventional and different from the more common statistical procedures used in this field, which readers might expect. However, it is also described in the main text (lines 146-160). We think it substantially improves the analysis and addresses the reviewers’ concern.

Overall, we have now done conventional differential expression analyses, Mann-Whitney tests as done by Hollis et al, added bootstrapping, heatmaps and now this novel resampling of replicate means. Each procedure gives essentially the same results, that follow our original interpretations of the data. We would like to think that this represents a constructive engagement with the suggestions (rather than ‘not really doing much about it’) and should be enough to reassure the reviewer that our results are robust.

In re-reading the Methods, I became worried that there might be unintended differences between virgin and non-virgin flies (Ln 413-416). I would like a clear and explicit statement about all the differences between virgin and non-virgin flies including age, handling, or any other differences they can think of.

- This seems to have resulted from us using the word ‘Virgin’ to apply to all females prior to the mating treatment, reasonably interpreted as potentially a distinct way of handling the Virgin group. We have clarified this (lines 429-435).

Minor comments:

Ln 108: What does “marginally” mean here. This is a bit of a loaded word often meant to imply $0.05 < p < 0.1$.

- Actually, this is one of the few changes in results following our re-filtering and this statement has become unnecessary.

Fig. 1&2. I could not find where the stars are explained in terms of specific p-value ranges.

- This has been added to the legends.

Ln 178. Please provide a brief explanation of how/why glutathione metabolism is linked to sexual selection. I have no idea and I suspect most readers won’t.

- We have elaborated on this statement (lines 190-192)

Reviewer #2 (Remarks to the Author):

The authors have addressed my previous concerns adequately and improved the manuscript. I have no further comments.

Reviewer #3 (Remarks to the Author):

Veltsos et al. have done a thorough job of addressing comments from all of the reviewers. In particular, the bootstrapping analysis of gene expression changes is an important statistical improvement. The integration of existing testis and ovary data, as well as the genomic distribution analysis (X vs. autosome), makes this study far more nuanced than the original Hollis et al study. The use of a baseline population for comparisons is a particular strength of the current study, as well as the choice of tissues and the novel data relating to post-copulatory expression. The depth of the analysis is now impressive and the complexity of the outcomes suggests that sexual antagonism and the evolution of sex-biased gene expression are perhaps far more complicated than many were beginning to assume. This is perhaps the most important contribution of the study and will hopefully motivate additional investigation. The importance of understanding genetic variation in the source population is discussed appropriately and is certainly something that is critical for future studies to take into consideration. I have a couple of additional suggestions that may be worth consideration prior to publication.

1. The changes in expression patterns are complex in nature and therefore somewhat difficult to keep straight when reading the manuscript. Might it be helpful to readers to move Fig S4 to the main text? This figure proved useful to me and is effective in summarizing the main trends.

- We have now done this, new figure 3

2. I appreciate the inclusion of the new testis and ovary analyses and it is interesting to see that the patterns remain after removal of male:female gonad biased genes. In my initial comment, I was thinking that this data could be used to estimate the extent to which non-gonadal sex biased genes contribute to the overall patterns observed. As would be expected, there are far more genes that are sex-biased in the abdomen dataset relative to the brain but we don't know the full extent to which this is accounted for by "reproductive" genes. It is quite possible that other tissues in the abdomen (including both other reproductive and non-reproductive tissues) might contribute meaningfully to sex biased genes expression and gene expression evolution. Does this make sense? It would be interesting to estimate this and determine what non-gonadal tissues might be responsive to shifts in sexual selection. Instead of comparing testis to ovary to exclude genes (as you have done), I was thinking of comparing testis and ovary to whole male/female expression to estimate gonad specificity and then determine if and to what extent somatic sex-biased genes respond to shifts in mating system. Sorry, I'm not sure I'm explaining this well.

- We agree that it would be interesting to separate the effect of reproductive tissues and other abdominal tissues. However, the data necessary for such a test are not ideal. We used the omnibus GSE52058 dataset based on reproductive tissues, which is used to identify

sex-biased genes in the associated publication. But this dataset is not informative about genes expressed in other 'non-reproductive' abdominal tissues because they are derived from dissected reproductive tissues. In order to separate abdominal-only from reproductive-only genes for D. pseudoobscura, we would need to use abdominal RNAseq data from yet another study, available in MODencode. This would then require combining data from three different studies, each obtained in different laboratories using different techniques. Frankly, we are reluctant to add this to the paper, though we do appreciate the reviewers' suggestion.

We did try using the more detailed D. melanogaster flyatlas data to specifically remove genes expressed in reproductive tissues from our abdominal data and the main pattern of masculinisation under monogamy remained unchanged. We decided not to include this as supplementary information because it would lack any information on species-specific genes.

Overall three analyses (removal of testes and ovary enriched genes – figure S1, removal of abdomen enriched genes– figure S2, flyatlas –shown below but not added to paper) have not affected the main patterns concerning directions of change under monogamy.

3. I'm curious about why the "correlation between sexes" analysis was restricted to only significantly differentially expressed genes. The pattern emerging from this study (and the hollis study) is small shifts in sex-biased expression optima across many genes and it would therefore be more informative to look for relationships across the full set of male and female biased genes. Perhaps a correlational analysis is not ideal in this case. One could simply ask if the directionality of change in males and females overlaps more than would expected by chance. This would be relative easy to test by simulation.

- We initially thought that restricting the comparison to genes found to be DE in our study would be better because this would minimise potential false positives due to minor non-significant variation across many thousands of genes (see the exchange with reviewer 1 about gene-based analyses). Nevertheless, new figure S4 now includes both all homologous genes, and those DE in our study. In fact, there are significant correlations across all genes. In female heads, the correlation is positive. In males it is somewhat surprisingly significantly

negative but the correlation coefficient is only -0.05. This is heavily influenced by outliers and becomes +0.06 if we are more rigorous and remove the top and bottom 1% of the data points. We do suspect that these are unlikely to be very biologically meaningful correlations involving so many non-significant changes in gene expression between different studies but mention them, with caveats.(lines 218-232)

4. Similarly, I'm concerned the genomic distribution analysis may suffer from the same sort of problem. Unique patterns of change for sex linked genes may be missed when focusing only on DE genes. Would it be possible to statistically compare the full distribution of gene expression changes between autosomes, XL and XR genes? Theory predicts differences in response between these classes of genes and it would be unfortunate to miss potentially informative patterns because the analysis focusses solely on FDR 10%.

- The test we presented was on whether the proportion of DE genes on sex-chromosomes and autosomes differed from expectations given the gene density on these chromosomes, so this test can only be applied to DE genes.

Alternatively, we can ask if on average the logFC changes due to treatment across all genes differs between the chromosomes. To test this we carried out an ANOVA on the magnitude of FC due to treatment for all genes included in each contrast, classified by whether they were on XL, XR, A2, A3, or A4 (this has to be done separately for each contrast as different numbers of genes were included in each contrast). There were significant differences by chromosome for 5 contrasts, but no consistent patterns. For example, for virgin female heads sex-linked genes showed smaller expression changes than autosomes and for only XL in male courted bodies did sex-linked genes show larger changes. But significant changes lacked any consistent or informative pattern. We would prefer not to add this to the published paper, it would require another lengthy supplement, which merely reinforces what we already say more succinctly from the previous analysis, that there is no suggestion of an overall greater response in sex-linked genes.

Reviewers' comments:

Reviewer #1 (Remarks to the Author):

I am content with the revisions.

One minor point: In the previous review I has requested that the authors provide a brief explanation of how/why glutathione metabolism is linked to sexual selection. In their response they said they had elaborated on this in lin 190-92 but I think this is the same as before (or maybe it is new but it is still not informative).

Reviewer #3 (Remarks to the Author):

Overall, I am pleased with the authors responses to my suggestions. However, after considering the previous comments by Reviewer #1 much more carefully, I feel obliged to comment in greater detail on the statistical methods. I apologize for not doing so adequately in my previous response.

It is absolutely clear that replication is at the population level and that a statistical approach based on the level of the gene will greatly inflate significance and is inappropriate. The fact that the EdgeR package has been cited highly seems irrelevant in this discussion, although it has proven to be a very useful and popular tool. EdgeR is dealing with overdispersion with the goal of identifying differentially expressed genes. The issue at hand is the distribution of gene expression changes between classes of genes. The fact that Hollis et al. used this approach does not in my opinion lend validation to its continued use in this study. The current study is superior to the Hollis study in several important facets and this should be extended to the statistical analyses. The results from the new simulation analysis generally confirm the trends already reported so they should be presented as the primary statistical support (however I do have some questions/concerns about the methodology; see below). I do not agree with the argument that the Hollis et al methodology should be presented to allow a more direct comparison between studies. The studies differ in many ways and a comparison of the overall trends does not require that identical (and what would appear to be inappropriate) analyses provide the basis for the main statistical results.

1. I agree with the authors that a simulation is likely to be the best solution. I do not, however, understand why the approach is based on swapping only half of the genes. What is the justification for this? It seems arbitrary. Given the underlying trends, this may actually make the test more conservative but it's not entirely clear from the description of the methodology. I believe that the null distribution should be derived from full Monte Carlo randomization across treatments and replicates for each gene. With 4 x 2 replication it will probably be necessary to do more than 1000 iterations. There is a large literature on permutation analyses to draw from. I'm also concerned about how means were generated, but I may be misunderstanding the approach. There is huge variation in RPKM between high and low expressed genes and highly expressed genes seem likely to dominate the calculation of averages. A geometric mean should be used so that each gene contributes an

equal amount to the overall directional change both for observed and simulated data. This should control for the contribution of those changes in highly expressed genes. The methodology should be included in the main methods section and not in supp. materials.

2. Returning to the issue of differentially expressed genes vs. directional differences in classes of genes, the new statistical discussion between lines 146-150 is confusing as written and seems to miss the point. This should be removed when the statistical support is derived from the new simulations, bootstrapping and heatmaps. Although I do think it is important to highlight why the MC simulation is used and how it improves on the methodology of Hollis et al. I don't think what is currently stated will be clear to most readers.

3. The discussion of the bootstrapping methodology is insufficient. Please provide more details and indicate whether different levels of subsampling were explored.

Responses to Reviewers' comments:

Reviewer #1 (Remarks to the Author):

I am content with the revisions.

One minor point: In the previous review I has requested that the authors provide a brief explanation of how/why glutathione metabolism is linked to sexual selection. In their response they said they had elaborated on this in line 190-92 but I think this is the same as before (or maybe it is new but it is still not informative).

We have elaborated this section again to explain the reported links between glutathione metabolism and sexual selection (lines 204-209).

Reviewer #3 (Remarks to the Author):

Overall, I am pleased with the authors responses to my suggestions. However, after considering the previous comments by Reviewer #1 much more carefully, I feel obliged to comment in greater detail on the statistical methods. I apologize for not doing so adequately in my previous response.

It is absolutely clear that replication is at the population level and that a statistical approach based on the level of the gene will greatly inflate significance and is inappropriate. The fact that the EdgeR package has been cited highly seems irrelevant in this discussion, although it has proven to be a very useful and popular tool. EdgeR is dealing with overdispersion with the goal of identifying differentially expressed genes. The issue at hand is the distribution of gene expression changes between classes of genes. The fact that Hollis et al. used this approach does not in my opinion lend validation to its continued use in this study.

The use of EdgeR is not irrelevant, because that package was used to calculate the adjusted FC differences for each gene using an appropriate measure of dispersion given the design. These are typically used to determine differential expression if that is the objective (as it usually is). Here, we are seeking to compare patterns of fold-change difference across gene classes, according to their sex-bias.

We are all in agreement now that simply treating each gene independently is inappropriate. We have revised our approach and now present two new analyses. One (we call "balanced randomisation") randomises the direction of changes detected between treatment groups. The

other (“standard randomisation”) resamples the 4 replicate libraries in each treatment group. We prefer the former, and present the second as a supplement. Both give the now familiar results.

The current study is superior to the Hollis study in several important facets and this should be extended to the statistical analyses. The results from the new simulation analysis generally confirm the trends already reported so they should be presented as the primary statistical support (however I do have some questions/concerns about the methodology; see below). I do not agree with the argument that the Hollis et al methodology should be presented to allow a more direct comparison between studies. The studies differ in many ways and a comparison of the overall trends does not require that identical (and what would appear to be inappropriate) analyses provide the basis for the main statistical results.

The Hollis *et al.* approach, while inappropriate, has been influential and we still believe that it should be presented, because this allows us to explain the issues raised throughout this review process and to therefore justify the improved analysis (we see people using this framework, for example, in talks and posters at the recent ESEB congress). We would therefore prefer it to remain part of the main text, with appropriate caveats, followed by the new analysis. If the editor disagrees, we would consider changing this, but we still think that this will be more logical to the reader, and note that the other reviewers were content with this.

1. I agree with the authors that a simulation is likely to be the best solution. I do not, however, understand why the approach is based on swapping only half of the genes. What is the justification for this? It seems arbitrary. Given the underlying trends, this may actually make the test more conservative but it's not entirely clear from the description of the methodology.

In our balanced randomisation, swapping half the genes is the best possible mixture of the high and low sexual selection treatments (to give a silly example, if we moved 100% it would be like swapping the column names and reversing the pattern). We effectively obtain a distribution of differences between groups that are as mixed as possible and compare that (null) distribution to our observed difference. Swapping 50% of the genes generates a test statistic distributed around zero.

I believe that the null distribution should be derived from full Monte Carlo randomization across treatments and replicates for each gene. With 4 x 2 replication it will probably be necessary to do more than 1000 iterations. There is a large literature on permutation analyses to draw from.

First, a full Monte Carlo approach is neither necessary nor optimal because it is used as an approximation when all possible randomised combinations are too numerous to obtain. Since we are randomising at the gene level, all possible combinations in our 4 x 2 design are directly sampled. As a result, our direct sampling is superior to a Monte Carlo partial randomisation. Second, we have increased the number of iterations from our previous randomisation procedure to 10,000.

Had we had more than 4 replicates per sexual selection treatment, and more possible combinations, a Monte Carlo approach would have been justified. To examine this to the extent possible given our data, we also randomised by sub-setting replicates in our “standard randomisation”, but because our replication is not high, we present this as supplementary information only. It is worth emphasising that the results of the different approaches give consistent outcomes. We believe that the approach we now implement in the main paper is the most powerful possible with our data.

I'm also concerned about how means were generated, but I may be misunderstanding the approach. There is huge variation in RPKM between high and low expressed genes and highly expressed genes seem likely to dominate the calculation of averages. A geometric mean should be used so that each gene contributes an equal amount to the overall directional change both for observed and simulated data. This should control for the contribution of those changes in highly expressed genes. The methodology should be included in the main methods section and not in supp. materials.

We agree that highly expressed genes potentially had a disproportionate influence in our previous analysis. Using the geometric mean, as suggested by the reviewer, would give equal weight to all genes. We believe this is still inappropriate, because genes with low expression level tend to be more variable, and should not influence the result as much as genes with a more consistent expression pattern, and hence greater confidence. The p-value from the EdgeR analysis takes into account the observed gene-specific variability in gene expression, using either the variability of all genes of similar expression level, or the variability of a particular gene, whichever is more conservative. As a consequence, the p-value from EdgeR incorporates information both on the extent of the difference between the sexual selection treatments (logFC) and weights individual genes based on their variability. We have therefore derived a new metric we call 'stLogFC' based on resampling the p-value from the EdgeR analysis. This is described in the main text and the code is provided. It involves quantiles, which are useful because they are less susceptible than means to long-tailed distributions and outliers, further addressing this concern.

2. Returning to the issue of differentially expressed genes vs. directional differences in classes of genes, the new statistical discussion between lines 146-150 is confusing as written and seems to miss the point. This should be removed when the statistical support is derived from the new simulations, bootstrapping and heatmaps. Although I do think it is important to highlight why the MC simulation is used and how it improves on the methodology of Hollis et al. I don't think what is currently stated will be clear to most readers.

We have tried to present the new analysis more clearly. We believe strongly that it is necessary to justify the new stdLogFC approach given what is already in the public domain, and the section in previous lines 146-150 was our attempt to do this. Everything has now been re-written but we reiterate that we believe that it is easiest to justify and explain our approach by presenting the original analysis, explaining the issues with that, then presenting the new, improved analysis. Otherwise authors familiar with studies in this area will not understand why we present this new, novel, methodology without justification.

3. The discussion of the bootstrapping methodology is insufficient. Please provide more details and indicate whether different levels of subsampling were explored.

We now extensively describe in detail the balanced randomisation test in the main text. We also describe the additional standard randomisation approach in the Supplementary Files, and we include the code for both analyses in the supplementary material.

REVIEWERS' COMMENTS:

Reviewer #3 (Remarks to the Author):

The authors have addressed my main concerns and I thank them for revisiting the statistics. I hope that they are correct that the discussion of the statistical issues during this review process will be helpful to the field and in future studies.